# Advanced Foot-And-Mouth Disease Vaccine Platform for Stimulation of Simultaneous Cellular and Humoral Immune Responses

**DOI:** 10.3390/vaccines8020254

**Published:** 2020-05-28

**Authors:** Min Ja Lee, Hyundong Jo, So Hui Park, Mi-Kyeong Ko, Su-Mi Kim, Byounghan Kim, Jong-Hyeon Park

**Affiliations:** Animal and Plant Quarantine Agency, 177 Hyeoksin 8-ro, Gimcheon-si, Gyeongsangbuk-do 39660, Korea; qncjgusehd@naver.com (H.J.); sohui33@korea.kr (S.H.P.); mkk80@korea.kr (M.-K.K.); beliefsk@korea.kr (S.-M.K.); kimbh61@korea.kr (B.K.)

**Keywords:** foot-and-mouth disease, vaccine, immunopotent, cellular and humoral immunity, protection

## Abstract

Currently available commercial foot-and-mouth disease (FMD) vaccines have various limitations, such as the slow induction and short-term maintenance of antibody titers. Therefore, a novel FMD vaccine that can rapidly induce high neutralizing antibody titers to protect the host in early stages of an FMD virus infection, maintain high antibody titers for long periods after one vaccination dose, and confer full protection against clinical symptoms by simultaneously stimulating cellular and humoral immunity is needed. Here, we developed immunopotent FMD vaccine strains A-3A and A-HSP70, which elicit strong initial cellular immune response and induce humoral immune response, including long-lasting memory response. We purified the antigen (inactivated virus) derived from these immunopotent vaccine strains, and evaluated the immunogenicity and efficacy of the vaccines containing these antigens in mice and pigs. The immunopotent vaccine strains A-3A and A-HSP70 demonstrated superior immunogenicity compared with the A strain (backbone strain) in mice. The oil emulsion-free vaccine containing A-3A and A-HSP70 antigens effectively induced early, mid-term, and long-term immunity in mice and pigs by eliciting robust cellular and humoral immune responses through the activation of co-stimulatory molecules and the secretion of proinflammatory cytokines. We successfully derived an innovative FMD vaccine formulation to create more effective FMD vaccines.

## 1. Introduction

Foot-and-mouth disease (FMD) is a highly contagious viral disease that typically affects cloven-hoofed livestock and leads to huge economic losses in the livestock industry due to its rapid transmission, severe reduction in animal productivity, and high mortality in newborn animals caused by myocarditis [1]. FMD is categorized as a category I livestock epidemic by the Act on the Prevention of Livestock Epidemics in the Republic of Korea and is managed by the World Organization for Animal Health (Office International des Épizooties; OIE). As such, FMD cases must be reported to the OIE [2]. Susceptible species include domestic ruminants, including cattle, water buffaloes, camels, sheep, goats, and pigs, and about 70 or more wild animal species. This disease is accompanied by a high fever and causes vesicle formation on the mouth, tongue, snout, nose, teats, hooves, and other hairless parts of the skin [3].

Foot-and-mouth disease virus (FMDV), the etiological agent underlying FMD, is a single-stranded, positive-sense RNA virus belonging to the *Aphthovirus* genus of the family *Picornaviridae*. The virus is classified into 7 distinct serotypes (A, O, C, Asia1, SAT1, SAT2, and SAT3) [4]. Viruses that share ≥85% nucleotide identity in the region of the FMDV genome corresponding to VP1 form a single serotype, and such classification, in most cases, is geographically restricted and termed a topotype. FMDV shows a high degree of genetic and antigenic variations, such that antibodies induced by one serotype cannot neutralize viruses of different serotypes, and vaccination does not offer cross-protection [5,6].

Therefore, to protect against each serotype and topotype, vaccine strains for each type must be developed and used. Additionally, field strains must be continuously cultured and adapted to maintain antibody titer potency. However, these methods are time-consuming, and objective data on the efficacy of vaccines containing these vaccine strains remain ambiguous. Given this background, it is necessary to design recombinant viruses using vaccine strains that are effective in terms of host protection and have abundant available information with regard to their function. This will allow the required vaccine strains to be rapidly developed and used when they are needed.

Although vaccination policy has been implemented to prevent and treat FMD [7], numerous drawbacks of commercially available FMD vaccines have been pointed out [7,8,9]. These drawbacks include (1) the long time required to establish a protective level of vaccine-mediated antibodies; (2) low and short-lived antibody titers and lower immunogenicity in pigs than in cattle; (3) requirement of regular and repeated vaccination at 4–6 months or 1 month intervals in cattle and pigs, respectively; (4) side effects including granuloma and abscess formation at the vaccination site, and safety problems caused by oil-based adjuvants in FMD vaccines when intramuscularly administered in pigs; (5) inability to induce cross-immunity or cross–protection within and between serotypes; (6) incomplete host protection with a humoral immune response alone; and (7) interference by maternally derived antibodies.

Therefore, to overcome the limitations of currently existing FMD vaccines and to increase FMD vaccine efficacy, innovative solutions are required, such as immunopotent FMD vaccine strains and the development of new non-oil adjuvants (emulsions, immunostimulants, etc.) for application in FMD vaccines. These immunopotent FMD vaccine strains and non-oil adjuvants will increase the immunogenicity of the antigen itself, induce long-term protective antibodies and memory response, and simultaneously elicit robust cellular and humoral immune responses.

In this study, immunopotent FMD vaccine strains A-3A and A-HSP70 were developed by spiking of the universal T cell FMDV epitope, 3A, or the HSP70 active site in the P1 region of FDMV, respectively, to increase the initial cellular immune response after vaccination and induce long-lasting memory responses. The purified antigens derived from these strains were used to prepare the vaccine formulation. To provide a new strategy for FMD vaccine formulation, mice and target animals (pigs) were vaccinated with these vaccines to evaluate the immunogenicity; induction of early, mid-term, and long-term immunity; and vaccine efficacy.

## 2. Materials and Methods

### 2.1. Recombinant Plasmid Preparation

The recombinant plasmid was prepared as described by Lee et al. [10]. The whole FMD-O1 Manisa virus genome (GenBank Accession No. AY593823.1) was amplified by PCR. The amplified O1-Manisa genome was inserted into a plasmid (pBluescript SK II) to prepare the pO-Manisa plasmid. In the prepared pO-Manisa, the gene that encodes the structural protein was substituted with the gene that encodes the structural protein from A-serotype FMDV A22/Iraq/24/64 (GenBank Accession No. AY593764.1) to prepare plasmids with the pOm-A22-P1. With the plasmids prepared as described above (pOm-A22-P1), the universal T cell epitope 3A sequence (GCAGCAATTGAATTCTTTGAGGGAATGGTGCATGACTCCATCAAG), which corresponds to the amino acid residue sequence (AAIEFFEGMVHDSIK), was inserted between the 453 and 454 base pair position (the 151 and 152 amino acid position) of the VP1 sequence. Then, 300 ng/μL of pOm-A22-P1 as PCR template, 1 μL of 10 pmole/μL primer 3A F (5′-GGAATGGTGCATGACTCCATCAAGGCGAGGGTCGCCGCTCAGCT-3′), and 1 μL of 10 pmole/μL primer 3A R (5′-CTCAAAGAATTCAATTGCTGCCGCGAGAGGCCCTAGGTCGC-3′) were used for the preparation of target plasmid by the same self-ligating method used in the previous study [10]. Figure 1A represents the schematic diagram of the final plasmid for A-3A.

For the preparation of A-HSP70, the heat shock protein (HSP) 70 epitope sequence (CAACCGTCGGTGCAGATCCAGGTCTATCAGGGGGAGCGTGAGATCGCCGCGCACAACAAG), which corresponds to the amino acid residue sequence (QPSVQIQVYQGEREIAAHNK), was inserted between the 453 and 454 base pair position of the VP1 sequence. Next, 300 ng/μL of pOm-A22-P1 plasmid, 1 μL of 10 pmole/μL primer HSP70 F (5′-GGGGAGCGTGAGATCGCCGCGCACAACAAGGCGAGGGTCGCCGCTCAGCT-3′), and 1 uL of 10 pmole/μL primer HSP70 R (5′-CTGATAGACCTGGATCTGCACCGACGGTTGCGCGAGAGGCCCTAGGTCGC-3′) were used. Figure 1B represents the schematic diagram of the final plasmid for A-HSP70.

PCR conditions were as follows: 10 μL of 5x Phusion HF buffer (Thermo Scientific, Waltham, MA, USA), 1 μL of 10 mM dNTP (Invitrogen, Carlsbad, CA, USA), 1 μL of 2 U/μL Phusion DNA polymerase (Thermo Scientific), and 35 μL of sterile distilled water were subjected to amplification for 25 cycles at 98 °C for 30 s, 98 °C for 10 s, 65 °C for 20 s, 72 °C for 2 min and 30 s, and a final cycle at 72 °C for 10 min. Next, 1 μL of DpnI (Enzynomics, Daejeon, Korea) was added to the 25 μL of PCR product and allowed to react in a 37 °C incubator for 1 h. Then, 35 μL of sterile distilled water, 5 μL of Ligation High (TOYOBO, Osaka, Japan), and 1 μL of 5 U/μL T4 polynucelotide kinase (TOYOBO, Osaka, Japan) were added to 4 μL of DpnI-treated product. The mixture was ligated in a 16 °C water bath for 1 h. Following ligation, the plasmid was transformed into 100 μL of DH5⍺ cells (Yeastern Biotech, Taipei, Taiwan) according to the manufacturer’s protocol. The transformed cells were smeared onto an agar plate containing ampicillin and incubated overnight at 37 °C. A colony was picked from the plate with a pipette tip and mixed with 18 μL of sterile distilled water, 1 μL of 10 pmol forward universal primer VP1 (5′-AGNGCNGGNAARTTTGA-3′), and 1 μL of 10 pmol/μL reverse universal primer VP1 (5′-CATGTCNTCCATCTGGTT-3′) in a colony PCR tube and subjected to 25 cycles of PCR amplification at 94 °C for 5 min, 94 °C for 30 s, 55 °C for 30 s, 72 °C for 1 min, and a final cycle at 72 °C for 5 min. In the above universal primer, N can represent any nucleotide. Five μL of PCR sample was mixed with 1 μL of 6x loading buffer (DYNE BIO, Gyeonggi-do, Korea) before being loaded onto an agarose gel. Then, 5 μL of 100 bp marker (DYNE BIO) was also loaded onto the gel. After electrophoresis at 100 V for 30 min, bands were assessed with Gel Doc. After assessment of the bands, 5 μL of PCR product was mixed with 2 μL of ExoSAP (Thermo Scientific) and amplified by PCR at 37 °C for 15 and 85 °C for 15 min. The insertion of the epitopes into VP1 was confirmed by full DNA sequencing. After confirming the sequence, the colony was placed in 200 mL of LB media containing ampicillin and incubated overnight at 37 °C with shaking. Midi prep (MACHEREY-NAGEL, Duren, Germany) was used to prepare the plasmid.

### 2.2. Immunopotent Recombinant FMD Virus Preparation

Recombinant FMD virus was recovered by transfection of BHKT7-9 (cell line that expresses T7 RNA polymerase) with the recombinant plasmid prepared above using Lipofectamine 3000 Reagent (Invitrogen, Carlsbad, CA, USA), followed by incubation for 2–3 days. Next, the prepared virus was passaged in fetal goat tongue (ZZ-R) cells or baby hamster kidney-21 (BHK-21) cells for viral proliferation.

### 2.3. Purification of Antigen (Inactivated Virus) from Recombinant Virus Presenting 3A- or HSP70-Epitope on the Surface

Purified antigen was prepared in BHK-21 cells infected with recombinant immunopotent FMDV A-3A and A-HSP70 constructed for the swift phenotype of P1 (referred sequence) by reverse genetics according to the method described by Lee et al., with modifications [10,11]. For viral infection, the culture medium was replaced with serum-free Dulbecco’s modified Eagle’s medium (DMEM; HyClone, Logan, UT, USA) and the cells were inoculated with the virus by incubating for 1 h at 37 °C in a 5% CO_2_ atmosphere. The extracellular viruses were then removed. Twenty-four hours post-infection, the viruses were inactivated by two treatments of 0.003 N binary ethylenimine for 24 h in a shaking incubator, followed by concentration with polyethylene glycol (PEG) 6000 (Sigma-Aldrich, St. Louis, MO, USA) [12]. The virus concentrate was layered on 15%–45% sucrose density gradients and centrifuged. After ultracentrifugation, the bottom of the centrifuge tube was punctured and 1 mL fractions were collected. The presence of FMDV particles in a sample of each fraction was confirmed by optical density using a lateral flow device (BioSign FMDV Ag; Princeton BioMeditech, Princeton, NJ, USA). Prior to use in field experiments, the pre-PEG treated supernatant was passaged through ZZ-R and BHK-21 cells at least twice to check that no cytopathic effects (CPE) occurred, thereby confirming the absence of any live virus in the supernatant.

### 2.4. Confirmation of Structural and Non-Structural Proteins Using Purified Antigens and Examination of 146S Particles Using TEM

Structural proteins (SPs) of purified antigen expression of cells infected with immunopotent recombinant FMDV A-3A and A-HSP70 were confirmed by rapid antigen kits (PBM kit, PBM Co Ltd., Princeton, NJ, USA), showing band formation for SPs and no band formation for non-structural proteins (NSPs) of FMDV. The virus particle (146S) was characterized by transmission electron microscope (TEM) imaging.

### 2.5. Confirmation of Immunogenicity in Mice

#### 2.5.1. Mice

The mouse experiment was conducted according to the method described in Lee et al. [11]. Age- and sex-matched wild-type C57BL/6 mice (6–7-week-old females) were purchased from KOSA BIO Inc. (Gyeonggi, Korea). All mice were housed in microisolator cages with ad libitum access to food and water in a specific pathogen-free (SPF) biosafety level 3 (ABSL3) animal facility at the Animal and Plant Quarantine Agency. All animals were allowed to adapt for at least one week before use in experiments. The housing room was set to a 12 h light/dark cycle, a temperature of about 22 °C, and a relative air humidity of about 50%. Studies were performed according to institutional guidelines and with approval from the Ethics Committee of the Animal and Plant Quarantine Agency (Accreditation number IACUC-2018-800 and IACUC-2019-185). 

#### 2.5.2. Vaccination and FMDV Challenge in Mice

To validate the immunogenicity and short-term immunity of purified antigen isolated from immunopotent FMDV A-3A and A-HSP70, and to verify their potential as a master seed virus for the development of an FMD vaccine, animal experiments were conducted according to the following strategy. Vaccine compositions used in the experiments are as follows: purified antigens were isolated from A-3A and A-HSP70 (15 μg/dose/mL, 1/10–1/640 dose for pigs), ISA 206 (Seppic, Paris, France; 50%, *w*/*w*), 10% Al(OH)_3_, and 15 μg/mouse Quil-A (InvivoGen, San Diego, CA, USA). Mice were vaccinated by intramuscular (I.M.) injection in the thigh muscle (0 days post-vaccination (dpv)) and challenged by intraperitoneal (I.P.) injection 7 dpv with FMDV (100 LD_50_ A/Malay/97, SEA topotype). Mice in the negative control group received an equal volume of phosphate-buffered saline (PBS, pH 7.0) administered via the same route. Survival rates and changes in body weight were monitored for up to 7 days post-challenge (dpc) to assess short-term immunogenicity (Figure 2A).

To evaluate the long-term immunity induced by test vaccines containing purified inactivated antigens isolated from A-3A and A-HSP70, respectively, to evaluate their efficacy in protecting the host against FMDV infection and to assess whether co-administration of A-3A and A-HSP70 simultaneously enhance cellular and humoral immune responses, mouse experiments were conducted according to the following strategy. Vaccine compositions used in the experiments were as follows: A (positive control, PC), A-3A, A-HSP70, or A-3A+A-HSP70 antigens (15 μg/dose/mL, A-3A or A-HSP 1/10 dose for pig, A-3A 1/20 dose+A-HSP70 1/20 dose for pig), without (*w*/*o*) oil emulsion, 10% Al(OH)_3_, and 15 μg/mouse Quil-A. Vaccination was performed twice, separated by a 35-day interval, and 100 μL vaccine (1/10 dose for pig) was administered by I.M. injection into the thigh muscle. Mice in the negative control group received an equal volume of phosphate-buffered saline (PBS, pH 7.0) administered via the same route. Blood samples were collected at 0, 7, 14, 28, 35, 56, 84, and 168 dpv. Antibody titers were determined by structural protein (SP) A ELISA, and virus neutralization (VN) titers were determined by VN tests from the sera. Animals were challenged by I.P. injection at 84 dpv or 168 dpv with FMDV (100 LD_50_ A/Malay/97, SEA topotype). Survival rates and changes in body weight were monitored for up to 7 dpc (Figure 3A).

### 2.6. Immunopotent Recombinant FMD Vaccine Strain-Induced Long-Term Immunity in Pigs

#### 2.6.1. Pigs

To evaluate the potential of A-HSP70 as a FMDV vaccine strain and to investigate its ability to induce cellular and humoral immune responses and long-term immunity, target animal experiments using pigs were conducted to generate preliminary data according to the method described by Lee et al. [11]. For the target animal experiment, FMD antibody-seronegative animals from a pig farm were used (the pigs were 10–12 weeks old). The pigs were divided into 2 groups: A (positive control, PC)-treated and an A-HSP70-treated groups (*n* = 5/group) (Appendix A).

Using the novel immunopotent FMD vaccine strains based on A-3A and A-HSP70 antigens, experiments were conducted in 10–12-week-old pigs (target animal) to evaluate the effects of emulsion-free (*w*/*o* oil emulsion) test vaccine including A-HSP70 antigen alone and A-3A+A-HSP70 antigen co-administration on the induction of early, mid-term, and long-term immunity. FMD antibody-seronegative pigs were used, and the animals were randomly divided into three groups (*n* = 5/group) (Figure 4A).

The animals were isolated in closed containments (ABSL3) during the study. After arrival in our ABSL, all animals were kept in the cage with ad libitum access to food and water and were used for the experiment after at least one week of adaptation. The housing room was set to a 12 h light/dark cycle, a temperature of about 22 °C, and a relative air humidity of about 50%. These studies were performed according to institutional guidelines, having received approval from the Ethics Committee of the Animal and Plant Quarantine Agency (Accreditation number IACUC-2018-800 and IACUC-2019-185).

#### 2.6.2. Immunization and Sampling

To verify the potential of purified antigens isolated from A-HSP70 as FMDV vaccine candidates and to test their ability to elicit robust cellular and humoral immune responses, antigens derived from A and A-HSP70 were used to formulate test vaccines. The vaccine composition for the test vaccines were as follows: 1 mL vaccine prepared as a single dose, which included 15 µg A antigen or A-HSP70 antigen, ISA 206 (50%, *w*/*w*), 10% Al(OH)_3_, and 150 µg Quil-A. Vaccination was performed twice with a 28-day interval, and 1 mL vaccine (1 dose for pig) was administered via a deep I.M. route in the animal neck. Pig blood samples were collected at 0, 14, 28, 42, 56, 70, and 84 dpv (Appendix A).

To examine the effect of purified antigens isolated from immunopotent FMD recombinant viruses to induce early, mid-term, and long-term immunity when administered alone or co-administered, the basic vaccine compositions used were as follows: A antigen (positive control, PC), A-HSP70 antigen (15 μg/dose/mL), or A-3A+A-HSP70 antigens (7.5 + 7.5 μg/dose/mL), *w*/*o* oil emulsion, 10% Al(OH)_3_, and 150 μg/pig Quil-A.

A-type FMD antibody-seronegative pigs were used. After the animals were given the first dose I.M. (0 dpv), a booster injection was administered at 28 dpv. Blood samples were collected at 0, 7, 14, 28, 42, 56, 70, and 84 dpv to examine antibody titers by SP A ELISA and VN titers. In addition, peripheral blood mononuclear cells (PBMCs) were isolated from whole blood to evaluate factors associated with cellular and humoral immune responses (Figure 5A).

Animals were monitored daily for body temperature, vaccination site symptoms, and appetite. Serum samples were stored at −80 °C until tests were performed.

### 2.7. Serological Assays

#### 2.7.1. ELISA for the Detection of Structural Protein Antibodies

To detect SP antibodies in the sera, PrioCHECK FMDV type A (Prionics AG, Switzerland) was used as described by Lee et al. [11]. Absorbance in the ELISA plate was converted to a percent inhibition (PI) value. When the PI value was 50% or above, the animals were considered antibody-positive.

#### 2.7.2. Virus Neutralization (VN) Test

A virus neutralization test was performed based on the guidelines set forth by the World Organization for Animal Health (OIE) manual [13], as described by Lee et al. [11]. Briefly, the sera were heat-inactivated at 56 °C for 30 min in a water bath. Cell density was adjusted to form a 70% monolayer, and 2-fold serial dilutions of sera samples (1:8–1:1024) were prepared. The diluted sera samples were then incubated with a 100-tissue culture infectious dose (TCID)_50_/0.5 mL homologous virus for 1 h at 37 °C. After 1 h, an LF bovine kidney (BK) cell suspension was added to all microplate wells. After 2–3 days, CPE was checked to determine the titers, which were calculated as Log_10_ of the reciprocal antibody dilution required to neutralize 100 TCID_50_ of the virus [14,15].

### 2.8. PBMC Isolation

Porcine PBMCs were isolated from whole blood of vaccinated pigs at the abovementioned specific time point (*n* = 5/group) according to the method derived by Lee et al. [11]. Whole blood (20 mL/donor) was independently collected in a BD Vacutainer heparin tube (BD, Becton, Dickinson and Company, Franklin Lakes, NJ, USA) and PBMCs were isolated using Ficoll-Paque^TM^ PLUS (GE Healthcare Bio-Sciences Corp., Piscataway, NJ, USA) gradient centrifugation. Residual red blood cells were lysed by treating them with ammonium–chloride–potassium (ACK) lysing buffer (Gibco, Carlsbad, CA, USA). The PBMCs were suspended in Dulbecco’s PBS without Ca^2+^ and Mg^2+^ (Gibco), supplemented with 2% fetal bovine serum (FBS) (Gibco), and counted using a volumetric flow cytometer (Miltenyi Biotec, Bergisch Gladbach, Germany). All cells were freshly isolated before use. No cryopreserved cells were used in any experiment. Purified PBMCs were then resuspended in RPMI1640 (Gibco) medium supplemented with 10% FBS (HyClone, Logan, UT, USA), 3 mM L-glutamine (Sigma-Aldrich, St. Louis, MO, USA), and 100 U/mL penicillin–streptomycin (Sigma-Aldrich).

### 2.9. RNA Isolation, cDNA Synthesis, and Quantitative Real-Time PCR

Total RNA from purified porcine PBMCs was extracted using TRIzol reagent (Invitrogen) and RNeasy Mini Kits (QIAGEN, Valencia, CA, USA). The cDNA was prepared by reverse transcription using a GoScript Reverse Transcription System (Promega, Madison, WI, USA) according to the manufacturer’s instructions. The synthesized cDNAs were amplified by quantitative-real-time PCR (qRT-PCR) on a Bio-Rad iCycler using iQ SYBR Green Supermix (BioRad, Hercules, CA, USA). Gene expression levels were normalized to *hprt* levels and presented as a relative ratio compared to control. The list of primers used in this study is presented in Appendix A.

### 2.10. Statistics

All quantitative data are expressed as mean ± SEM, unless otherwise stated. In the case of experiments for host protection against FMDV challenge, the negative control group (PBS-treated group) and the PC group (A (backbone strain) treated group) were compared with the experimental group. For experiments testing vaccine-mediated immune responses, the PC group and the experimental group were compared. Parametric tests were used to compare the groups. Between groups, statistical significances were assessed using two-way ANOVA followed by Bonferroni post-hoc test and one-way ANOVA followed by Tukey’s post-hoc test (*^, #^
*p* < 0.05; **^, ##^
*p* < 0.01; ***^, ###^
*p* < 0.001). Parametric tests were used to compare the groups. Survival curves were built using the Kaplan–Meier method and differences were analyzed using log-rank sum tests. GraphPad Prism 5 (GraphPad, San Diego, CA, USA) software was used for all statistical analyses.

## 3. Results

### 3.1. Identification of Candidate Immunopotent Materials, Including T Cell Epitope and Specific Target Genes that Stimulate Cellular and Humoral Immune Responses, and Preparation of FMD Vaccine Strains

Immunopotent FMD vaccine strains were developed by reverse genetics using the P1 backbone of the previously developed A22-R (A22 P1 replacement in the O1 Manisa infectious cDNA) strain [10,16].

Considering the time required for a B cell-mediated increase in antibody levels after vaccination, the universal T cell epitope 3A (15 a.a.) and the active site (20 a.a) of the 70-kDa heat shock protein (HSP70) were selected as the immunopotent candidate materials for simultaneous induction of cellular and humoral immune responses. The universal T cell epitope 3A can increase the initial T-cell-mediated cellular immune response after vaccination, and HSP70 serves as a linker between innate and adaptive immune responses, as well as induces long-term immunity and memory response [17,18,19]. Thus, immunopotent FMD vaccine strains were prepared by inserting the T cell epitope 3A and the epitope of long-lasting immune molecule gene HSP70 into the VP1 encoding region of A22 P1 backbone (Figure 1A,B).

### 3.2. Production and Purification of Antigens Using Immunopotent FMD Vaccine Strains

The prepared immunopotent FMDVs A-3A, A-HSP70, and FMDV A were used to produce antigens, which were then purified. The amount of antigen was measured and a screening test with field strains was performed with a PBM rapid test kit. Results confirmed SP band formation even with a very small 2.34 ng dose (1/640 dose) and no NSP band formation, indicating that the antigens were distinguishable from the field strains (Appendix A). The antigens (146S particles) purified using a sucrose gradient were analyzed with TEM, and the results are shown in Appendix A. These antigens were used as test vaccine antigens.

### 3.3. Evaluation of the Immunogenicity of Immunopotent FMD Vaccine Strains (A-3A and A-HSP70) in Experimental Mice

To (1) determine the immunogenicity of the purified antigens derived from the FMD virus A-3A, which is coupled with T cell epitope 3A to enhance cellular immune response, and FMD virus A-HSP70, which contains HSP70 to intensify long-term immunity; and to (2) evaluate their potential as a master seed virus (MSV) for the FMD vaccine strain and their protective effect against FMDV in mice, experiments were conducted according to the strategies illustrated in Figure 2A,D. Mice were vaccinated I.M. at 0 dpv and received an I.P. injection of FMDV (100 LD_50_ of A/Malay/97, SEA topotype) at 7 dpv. The survival rates (Figure 2B,E) and changes in body weight (Figure 2C,F, Appendix A) of the mice were monitored for up to 7 dpc.

Results showed that the survival rate of the mice vaccinated with the test vaccine containing A-3A antigen coupled with T cell epitope 3A was 100% at 1/10 and 1/40 doses, 80% at 1/160 dose, and 20% at 1/640 dose (32 protective dose (PD)_50_ when calculated as the mouse dose (one-tenth the dose for pigs equals 1 dose for mice)). Weight changes in these mice were also less than 10% at 1/10, 1/40, and 1/160 doses. The survival rate of the mice vaccinated with vaccine containing purified antigen derived from recombinant FMDV A-HSP70 designed for long-term immunity was 100% at 1/10 dose, 80% at 1/40 dose, and 0% at 1/160 and 1/640 doses (6 PD_50_ when calculated as mouse dose). However, a tendency towards delayed death was observed compared with the negative control group. Almost no weight change was observed with the 1/10 and 1/40 doses. Mice in the naïve control group (unvaccinated and unchallenged group) showed complete survival and normal increase in BW up to 7 dpc (data not shown). The results from this challenge study are encouraging, suggesting that both immunopotent recombinant FMD vaccine derived from A-3A and A-HSP70 might grant protection from experimental infection in mice, which is further validated later in this study.

### 3.4. Preliminary Evaluation of the Ability of Immunopotent FMD Vaccine Strain A-HSP70 to Induce Immunogenicity and Memory Response in Target Animals (Pigs)

To examine the efficacy of the purified antigen derived from the A-HSP70 to induce a memory response and long-term immunity in the target animal (pigs), preliminary experiments were conducted using field pigs. FMD antibody-seronegative, 10–12-week-old pigs were vaccinated twice I.M. with 1 mL vaccine at a 28-day interval (0 dpv, 28 dpv). Pig blood samples were collected at 0, 14, 28, 42, 56, 70, and 84 dpv and used for serological analysis. The detailed study strategy is shown in Appendix A.

The pigs vaccinated with test vaccine containing the immunopotent recombinant FMDV A-HSP70 antigen had significantly increased antibody titers measured by SP A ELISA (*p* < 0.01, Appendix A) and VN titers (*p* < 0.05, Appendix A) starting at 14 dpv. Following a booster injection at 28 dpv, significantly higher antibody and VN titers were maintained until 84 dpv compared with the PC group administered test vaccine containing A antigen (*p* < 0.001).

### 3.5. Efficacy of Non-Oil (Emulsion-Free) Test Vaccines Containing Immunopotent FMD Vaccine Strains (A-3A and A-HSP70) in Inducing Early, Mid-Term, and Long-Term Immunity and Memory Response in Mice and Pigs

Non-oil (emulsion-free) test vaccines were prepared using purified antigens derived from A-3A and A-HSP70. Then, the ability of these immunopotent FMD vaccine strains to induce immune and memory responses in mice was investigated. In addition, to enhance the induction of the initial immune response after vaccination, the A-HSP70 vaccine strain, which is more specialized for the induction of a long-term immune response, was co-administered with the purified antigen isolated from the A-3A vaccine strain. We then assessed whether these immunopotent vaccine strains could effectively induce early, mid-term, and long-term immunity in the host (Figure 3).

The antigens used consisted of immunopotent FMDV A-3A antigen and A-HSP70 antigen. Mice received the first dose of test vaccines by I.M. injection at 0 dpv and the second dose at 35 dpv. Following the first dose, blood samples were collected at 0, 7, 14, 28, 56, 84, and 168 dpv (Figure 3A) for serological analysis, including SP A ELISA (Figure 3B) and VN assays (Figure 3C). Mice received an I.P. injection of FMDV (100 LD_50_ A/Malay/97, SEA topotype) at 84 dpv or 168 dpv, and their survival rates (Figure 3D,F) and changes in body weight (Figure 3E,G) were monitored for up to 7 dpc.

Figure 3B,C demonstrate high antibody titers measured by SP A ELISA and VN titers in the A-3A alone administration, A-HSP70, and A-3A co-administration (A-3A+A-HSP70) groups following the first dose until 28 dpv. Following the second dose, high antibody levels and VN titers were maintained in the A-HSP 70 alone and A-3A+A-HSP70 groups from 58 to 168 dpv.

In FMDV (100 LD_50_ A/Malay/97, SEA topotype) challenge experiments, the A-3A alone, A-HSP70 alone, and A-3A+A-HSP70 groups all had a 100% survival rate at 84 dpv (Figure 3D) and 168 dpv (Figure 3F). No weight change was observed at 84 dpv (Figure 3E, Appendix A), and almost no weight change was observed at 168 dpv (Figure 3G, Appendix A). In contrast, the PC group administered naïve A antigen had a 40% protection rate upon FMDV challenge at 84 (Figure 3D) and 168 dpv (Figure 3F).

In addition, experiments were conducted to evaluate the long-term immunity and memory response of the emulsion-free (*w*/*o* oil emulsion) test vaccine containing novel immunopotent FMD vaccine strains, A-HSP70, and A-3A antigens in the target animal (pigs). Type-A FMDV antibody-seronegative pigs were used. Following the first dose given I.M. (0 dpv), a booster injection was given at 28 dpv and blood samples were collected at 0, 7, 14, 28, 42, 56, 70, and 84 dpv for the examination of antibody titers by SP A ELISA and VN titer. In addition, PBMCs were isolated from whole blood to determine the cellular and humoral immune response-related factors at 0, 14, 28, and 56 dpv (Figure 4A).

As shown in Figure 4B,C, antibody titers measured by SP A ELISA and VN titers were significantly higher in the A-3A+A-HSP70 group than in the A or A-HSP70 alone groups at 7 dpv (*p* < 0.001). Following the booster injection at 28 dpv, the A-HSP70 alone and A-3A+A-HSP70 groups both showed significantly higher values than A alone group (*p* < 0.001), while no significant differences were found between the A-HSP70 alone and A-3A+A-HSP70 groups.

### 3.6. Changes in the Expression of Co-Stimulatory Molecules and Cytokines Induced by Non-Oil (Emulsion-Free) Test Vaccine Containing the Immunopotent FMD Vaccine Strains A-3A and A-HSP70

Porcine PBMCs were isolated from whole blood collected from pigs vaccinated with non-oil test vaccines containing immunopotent FMD vaccine strains (A-3A and A-HSP70) at 0, 14, 28, and 56 dpv for RNA preparation (Figure 4A). Then, qRT-PCR was used to validate changes in gene expression patterns of co-stimulatory molecules and cytokines. The gene expression levels of the co-stimulatory molecules CD40, CD80, CD86, CTLA-4, major histocompatibility complex (MHC) class I, MHC class II, and cytokines IFNα, IFNβ, IFNγ, IL-12p40, and IL-17A were examined (Figure 5A–K). Notably, co-stimulatory molecules and expression levels of CD80 (Figure 5B) in the A-3A+A-HSP70 group increased more than 10-fold compared with the A alone group (*p* < 0.001), and were significantly higher than in the A-HSP70 alone group (*p* < 0.001) at 14 dpv. Gene expression of CD40 (Figure 5A) and CD86 (Figure 5C) were higher in the A-3A+A-HSP70 group compared to the A group (*p* < 0.01, and *p* < 0.05), and there was also a significant difference observed between the A-3A+A-HSP70 group and the A-HSP70 group at 28 and 56 dpv (*p* < 0.001, *p* < 0.05). Later, gene expression of MHC class I (Figure 5E) and MHC class II (Figure 5F) significantly increased in the A-3A+A-HSP70 group compared to the A group, similar to the expression of the abovementioned co-stimulatory molecules at 56 dpv (*p* < 0.001). In addition, the difference in gene expression between the A-3A+A-HSP70 group and the A-HSP70 group was also significant (*p* < 0.001). The gene expression level of CTLA-4 (Figure 5D) was increased in the A-3A+A-HSP70 group at 14 dpv, but was higher in the A group at 56 dpv. Overall expression was lower than other co-stimulatory molecules.

The gene expression levels of cytokines were higher in the A-3A+A-HSP70 group compared to the A alone group or A-HSP70 alone groups, and the gene expression levels of IFNα (Figure 5G), IFNβ (Figure 5H), IL-12p40 (Figure 5J), and IL-17A (Figure 5K) were the highest at 14 dpv (*p* < 0.001), slightly reduced at 28 dpv (*p* < 0.01, *p* < 0.05), and slightly increased again at 56 dpv *(p* < 0.01, *p* < 0.05). The gene expression level of IFNγ (Figure 5I) was also higher in the A-3A+A-HSP70 group compared to the A and A-HSP70 group at 14 and 28 dpv, but the differences between the groups were not significant at 56 dpv.

## 4. Discussion

Vaccination is used as a primary control strategy for FMD [7]. After a large outbreak of FMD, a nationwide vaccination policy was initiated in December 2010 in South Korea, calling for the mass production of vaccines in preparation for a large-scale vaccination scheme. FMD vaccines generally consist of inactivated vaccines in the form of a double-oil emulsion, and most vaccines use oil-based emulsion as adjuvants. All FMD vaccines focus on humoral immune responses linked to B cell-mediated increases in antibody levels. IgM antibodies are produced 3 to 4 dpv, peaking between 10 to 14 dpv and decreasing thereafter. IgG antibodies, as important neutralizing antibodies, are produced 4 to 7 dpv, reaching their maximum levels 3 weeks post-vaccination [20,21]. In contrast, in actual FMDV infections, virus concentrations in the blood reach their highest levels around day 2–3, and clinical symptoms are most severe around day 3.5–5 [22]. Therefore, protecting the host by stimulating the innate immune response and inducing the cellular immune response during the early stages of an infection is of utmost importance. In particular, clinical FMD is more severe in pigs and infected pigs are actually at a higher risk of spreading the virus because they can excrete a large amount of airborne virus [23]. However, clinical studies related to FMD have focused more on identifying the pathogenesis, enhancing vaccine efficacy, and increasing the effectiveness of vaccines in cattle rather than in pigs.

Currently, FMD vaccine studies using an inactivated vaccine, modified virus inactivated vaccine (DIVA vaccine) [24], viral vector vaccine [25], adenovirus [26,27,28], virus-like particle (VLP) vaccine (bacterial) [29], baculovirus [30,31], plant-produced empty capsids [32,33], peptide vaccine (Poly(I:C)) [34], CpG [35], plant based recombinant vaccine (alfalfa [36], chloroplast of tobacco [37]), RNA vaccine [38], DNA vaccine (cDNA [39,40], electroporation [41], T cell [42,43,44], B cell [45,46] epitope, APC targeting [47]), live attenuated vaccine [48,49,50,51], and antiviral agent [52,53] are underway. However, no high-quality FMD vaccines that surpass the efficacy of the inactivated vaccine have been successfully developed to date.

Therefore, a new strategic approach is urgently needed to ensure effective protection against FMD. The present study tested the immune cell-stimulating epitopes and target molecules for the development of an immunopotent FMD vaccine strain (Figure 1).

B cell activation and antibody production are generally associated with CD4^+^ T cell-mediated lymphocyte proliferation. Several T cell epitopes that are frequently recognized by host lymphocytes have been identified in FMDV proteins [17]. One of the T cell epitopes located in residues 21 to 35 of the non-structural protein (NSP) 3A is effectively recognized by pig lymphocytes [17,18]. It has been shown *in vitro* that a universal FMDV T cell epitope, when combined with the B cell antigenic site VP1, which is capable of eliciting significant levels of serotype-specific activity, can induce adequate cooperation of T helper cells (Th cells) and increase host protection against the pathogen [54,55]. Th1-type CD4^+^ T cells eliminate intracellular pathogens directly by inducing a cytokine response, such as the expression of IFNγ and TNFα, or indirectly by macrophage (MΦs) activation and CD8^+^ T cell differentiation [56]. In addition, Th2-type CD4^+^ T cells produce cytokines including IL-4, IL-5, and IL-13, and these cytokines have the ability to directly remove extracellular pathogens [57]. The amino acid sequence of 3A is conserved across FMDV serotypes A, O, and C, and shows limited variability between the 7 FMDV serotypes [18]. Therefore, we expect that the insertion of the T cell epitope 3A sequence as an immunopotent molecule into the FMD virus will elicit a CD4^+^ T cell response and promote cytokine secretion, including IFNγ, TNFα, and IL-17A. These cytokines will re-stimulate APCs, including dendritic cells (DCs), MΦs, and monocytes, leading to the secretion of inflammatory cytokines such as IL-12p40 and the polarization of a subset of T cells, while simultaneously stimulating MHC class II-TCR interaction, which will re-present antigens to T cells and eventually boost cellular immune responses.

HSP70 is referred to as a “chaperokine” to highlight their chaperone functions and cytokine-like effects. It serves as a linker between innate and adaptive immune responses [19]. In addition, HSP70, through endogenous signaling, is involved in the maturation, differentiation, and induction of recruitment of natural killer cells and their cytotoxicity [58], and increases the expression of proinflammatory cytokines, chemokine, and co-stimulatory molecules in APCs, such as DCs, monocytes, and MΦs [59]. APCs stimulated by HSP70 activate the NFκB pathway [60] and elicit Th1 and strong MHC class-I-restricted CD8^+^ T cell (CTL)-specific responses [61,62]. HSP70 acts as a chaperone for antigenic peptides in the endoplasmic reticulum and cytoplasm, and is involved in the processing and presentation of antigens, while playing a role in regulating immunity and tolerance [63]. FMDV infection sharply reduces the expression of MHC class I on the surface of susceptible cells, while FMDV-infected cells impair the presentation of viral peptides to CTLs, thereby facilitating virus escape from the host CTL response [64,65]. Therefore, we expect that continuous stimulation of CTL by spiking the FMDV surface with HSP70 will play an important role in host protection against the infection.

In this study, immunopotent FMD vaccine strains A-3A and A-HSP70 were shown to have superior immunogenicity in mice compared to the naïve A strain, which was used as the backbone (Figure 2). Furthermore, field experiments revealed the potential of A-HSP70 to enhance long-term immunity in the target animal (pigs) (Appendix A). Despite many claims that it is difficult to elicit an effective immune response in hosts with FMD vaccines that do not use oil emulsion, particularly in pigs, we confirmed that the administration of oil emulsion-free (non-oil emulsion) test vaccines containing purified antigens isolated from the immunopotent FMD vaccine strains effectively induce early, mid-term, and long-term immunity and memory response in mice (Figure 3). We also demonstrated that these emulsion-free (non-oil emulsion) test vaccines elicit robust humoral immune responses in the target animal (pigs). When the antigens were co-administered to harness the specialized advantage of each immunopotent FMD vaccine strain and to maximize their synergistic effect in improving the immune response, the antibody titers measured by SP A ELISA and VN titers were found to be high at all-time points following vaccination, from early to late stages (Figure 4).

With respect to gene expression levels of co-stimulatory molecules, significant differences in the expression levels of CD80, CD86, CD40, MHC class I, and MHC class II were found in the A-3A+A-HSP70 group compared with A group or A-HSP70 group. CD80 (B7-1) and CD86 (B7-2) are Ig superfamily members that interact with stimulatory CD28 or inhibitory CTLA-4 on T cells [66], and play crucial roles in the initiation and maintenance of the inflammatory response. These classical co-stimulatory molecules regulate T cell and B cell activation and the secretion of antibodies from B cells [67,68].

In particular, CD80 is transiently expressed on the cell surface of activated B cells, MΦs, DCs, and monocytes. Interestingly, CD80 enhance the memory response of CTLs. CD40, a member of the TNF receptor family, is among the most important co-stimulatory molecules involved in the T cell-B cell immune synapse. CD40 triggers an important signal for many aspects of B cell activation through its interaction with CD40L (CD154). CD40 deficiency leads to a defect in isotype switching, germinal center formation, and memory B cell response [69,70].

Antigen presentation by MHC proteins is essential for inducing adaptive immunity. MHC class I and MHC class II proteins play a pivotal role as linkers in the adaptive immune response. These two protein classes commonly serve to present peptides on the cell surface for recognition by T cells. The complex of immunogenic peptide and MHC class I is present in nucleated cells and is recognized by cytotoxic CD8^+^ T cells.

On the other hand, the presentation of peptide and MHC class II complexes by antigen-presenting cells such as DCs, MΦs, or B cells can activate CD4^+^ T cells, leading to the coordination and regulation of effector cells [71].

Based on this study’s findings that the gene expression of co-stimulatory molecules was significantly increased in the A-3A+A-HSP70 immunized group, but not the A-HSP70 or A immunized groups, it appears that these immunopotent vaccine strains, especially A-3A, effectively activate APCs, T cells, and B cells, eliciting robust innate and adaptive immune responses. We expect that these strains will confer strong protection to the host against FMDV infection through a CTL memory response upon infection with the pathogen. Further studies need to be conducted to further verify the cellular immune response and reveal the roles and mechanisms underlying the immune signaling pathway of A-3A and A-HSP70, respectively.

In addition, co-administration of these immunopotent vaccine strains elicits significant expression of proinflammatory IFNα, IFNβ, IFNγ, IL-12p40, and IL-17A. Based on the finding that co-administration increased the expression of (i) IFNα and IFNβ, both of which have antiviral characteristics; (ii) IFNγ, which is expressed in CD4^+^ T cells and CD8^+^ T cells; (iii) IL-12p40, which is primarily expressed in DCs and MΦs; and (iv) IL-17A, which is expressed in Th17 cells and γδ T cells, these immunopotent vaccine strains seem to effectively clear pathogens by recruiting neutrophils, MΦs, and NK cells to the infected area during the early stage of a FMDV infection (Figure 5).

## 5. Conclusions

Overall, immunopotent FMD vaccine strains A-3A and A-HSP70 simultaneously elicited strong cellular immune responses and long-lasting humoral immune responses in pigs, even as emulsion-free (non-oil emulsion), purified, inactivated antigen vaccines. We expect that these strains will be able to effectively protect the host against FMDV infection, provide innovative means of reducing local side effects, such as oil-emulsion-induced granulomas and fibroblasts at the vaccination site, and contribute to the establishment of a next-generation FMD vaccine for pigs.

## Figures and Tables

**Figure 1 vaccines-08-00254-f001:**
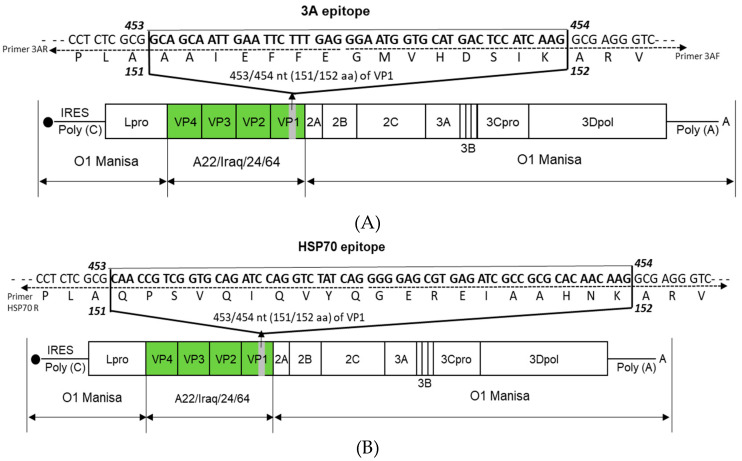
Construction of the immunopotent foot-and-mouth disease virus (FMDV) vaccine strains A-3A and A-HSP70. (**A**,**B**) represent: (**A**) A-3A; (**B**) A-HSP70. FMDV T cell epitope 3A (the 15 amino acid residues in VP1) (**A**) and enhanced long-term immunity protein HSP70 (the 20 amino acid residues of VP1) (**B**) were selected as target molecules for preparation of immunopotent virus. The A22 (A22-R) P1, where the P1 region of O1 Manisa is substituted with A22 P1, was used as the backbone for the preparation of immunopotent FMD vaccine strains. The detailed strategy is presented in the Materials and Methods.

**Figure 2 vaccines-08-00254-f002:**
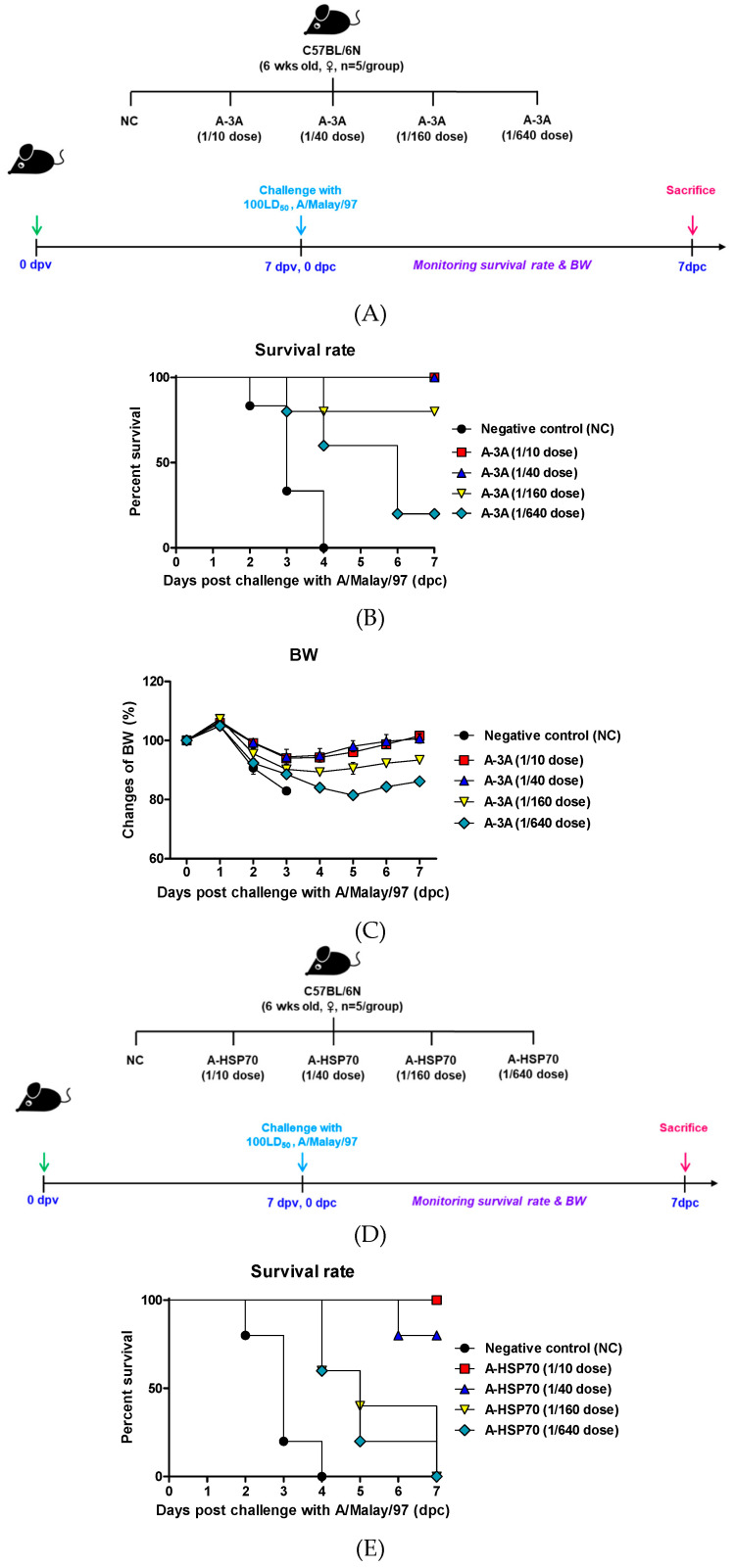
A-3A- and A-HSP70-mediated vaccine efficacy and protective effects in mice. (**A**–**F**) represent: (**A**–**C**) A-3A; (**D**–**F**) A-HSP70. C57BL/6 mice were administered the test vaccine at 1/10, 1/40, 1/160, 1/640 doses of A-3A or A-HSP70 antigen for cattle or pig use, ISA 206 (oil-based emulsion, 50%, *w/w*), 10% Al(OH)_3_, and 15 µg Quil-A. A negative control (NC) group was injected with the same volume of PBS. The test vaccines were injected intramuscularly into mice that were later challenged with FMDV (100 LD_50_ A/Malay/97) at 7 dpv. The survival rates and body weights were monitored for 7 dpc. Experimental strategy (**A**,**D**); survival rates post-challenge with A/Malay/97 (**B**,**E**); and changes in body weight post-challenge with A/Malay/97 (**C**,**F**). The data represent the mean ± SEM of triplicate measurements (*n* = 5/group). Statistical analyses were performed using two-way ANOVAs with a Bonferroni correction and a one-way ANOVA followed by a Tukey’s post-hoc test. ns, not significant. Statistical analyses are summarized in Appendix A.

**Figure 3 vaccines-08-00254-f003:**
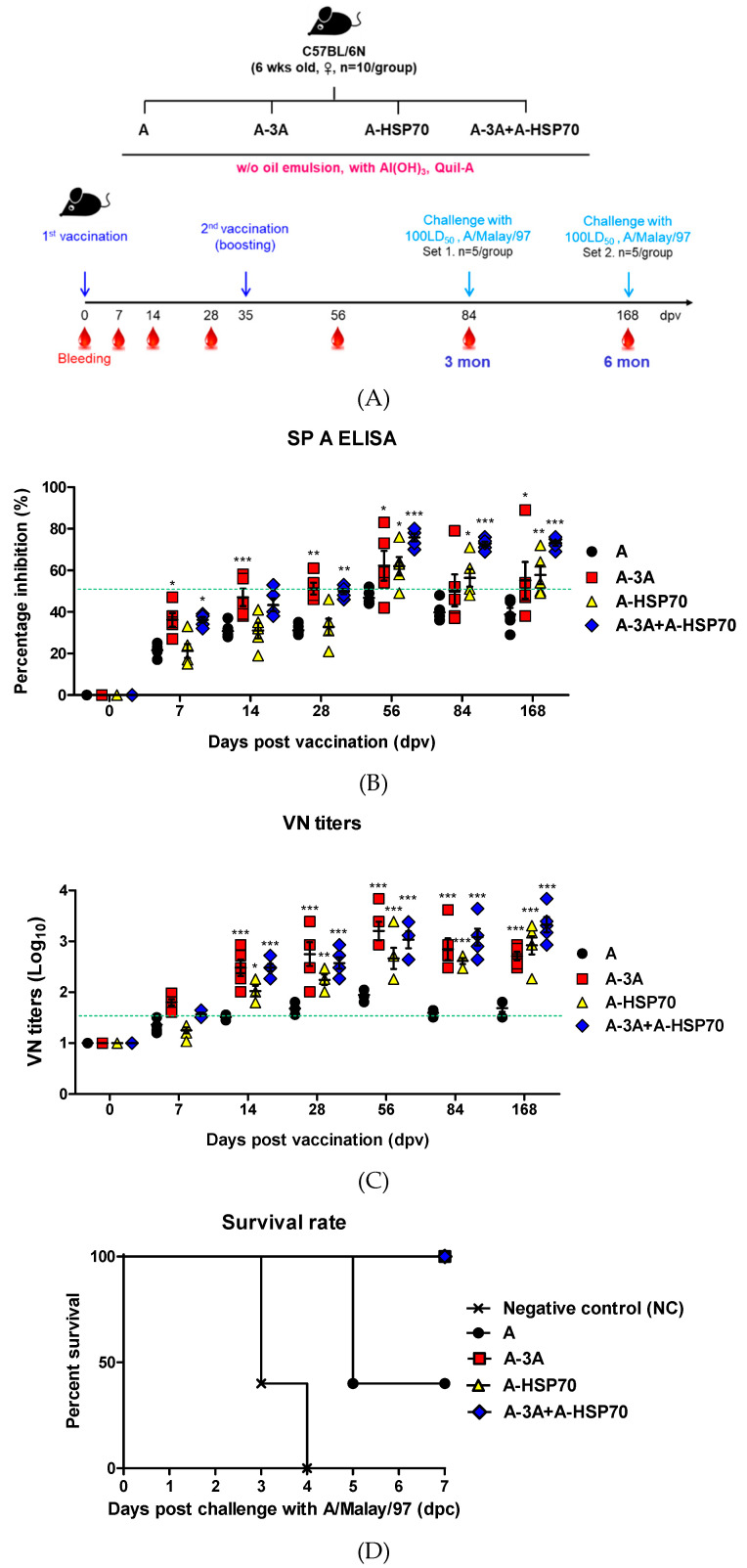
A-3A and A-HSP70 mediated a long-lasting memory response in mice. C57BL/6 mice were administered with oil emulsion-free test vaccines, including A-3A antigen alone, A-HSP70 antigen alone, or combined A-3A and A-HSP70 antigens based on their positive control composition. The negative control group received PBS in the same volume as the vaccine. A positive control group received 1.5 μg (1/10 dose for cattle and pig use) of naïve A antigen, without (*w*/*o*) oil emulsion, 10% Al(OH)_3_, and 15 μg Quil-A. Briefly, vaccination was performed twice at a 35-day interval. The mice were vaccinated intramuscularly in the thigh muscle. Later, at 84 dpv or 168 dpv, the mice were challenged with FMDV (100 LD_50_ A/Malay-97, SEA topotype) by intraperitoneal injection. Survival rates and body weights were monitored for 7 dpc. In addition, serum sampled from mice at 0, 7, 14, 28, 56, 84, and 168 dpv was analyzed via SP A ELISA and VN assay. (**A**–**G**) represent: (**A**) the strategy for this study; (**B**) antibody titers by SP A ELISA; (**C**) VN titers; (**D**) survival rate at 84 dpv challenge; (**E**) changes of body weight at 84 dpv challenge; (**F**) survival rate at 168 dpv challenge; (**G**) changes of body weight at 168 dpv challenge. The data represent the means ± SEM of triplicate measurements (*n* = 5/group). Statistical analyses were performed using two-way ANOVA with Bonferroni correction and a one-way ANOVA followed by a Tukey’s post-hoc test. * *p* < 0.05; ** *p* < 0.01; *** *p* < 0.001; ns, not significant. Statistical analyses are summarized in Appendix A.

**Figure 4 vaccines-08-00254-f004:**
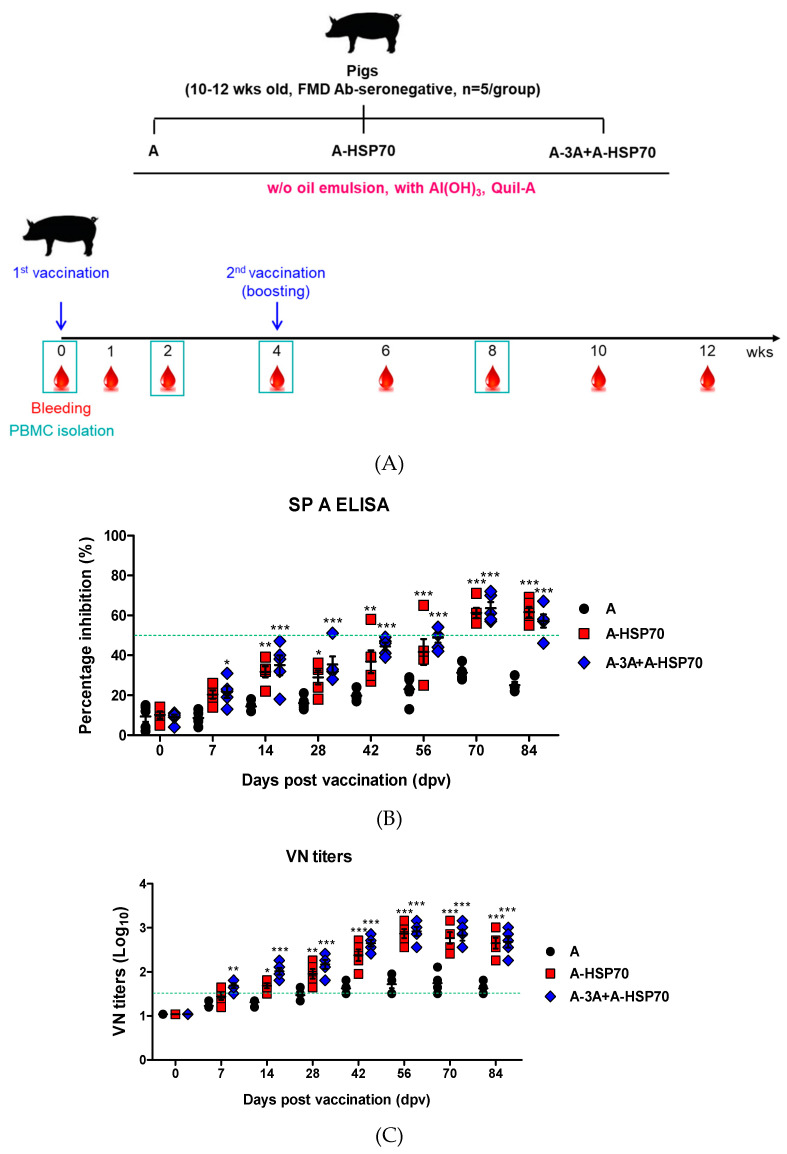
A-3A and A-HSP70 mediated-immune response in pigs. For the pig experiments, FMD antibody-seronegative animals were used (the pigs were 10–12 weeks old). Pigs were divided into 3 groups (*n* = 5/group). Pigs were administrated oil emulsion free-test vaccines, including A-HSP70 antigen alone, or combined A-3A and A-HSP70 antigens based on their positive control composition. A positive control group of pigs received 15 µg (1 dose for cattle and pig use) naïve A antigen, without (*w*/*o*) oil emulsion, 10% Al(OH)_3_, and 150 µg Quil-A. The vaccination was performed twice at a 28-day interval, and 1 ml vaccine (1 dose) was injected via a deep intramuscular route at the animals’ necks. Blood samples were collected at 0, 7, 14, 28, 42, 56, 70, and 84 dpv in pigs for the serological assays and at 0, 14, 28, and 56 dpv for PBMC isolation. (**A**–**C**) represent: (**A**) the strategy for this study; (**B**) antibody titers by SP A ELISA; (**C**) VN titers. The data represent the means ± SEM of triplicate measurements (*n* = 5/group). Statistical analyses were performed using two-way ANOVA with Bonferroni correction. * *p* < 0.05; ** *p* < 0.01; *** *p* < 0.001.

**Figure 5 vaccines-08-00254-f005:**
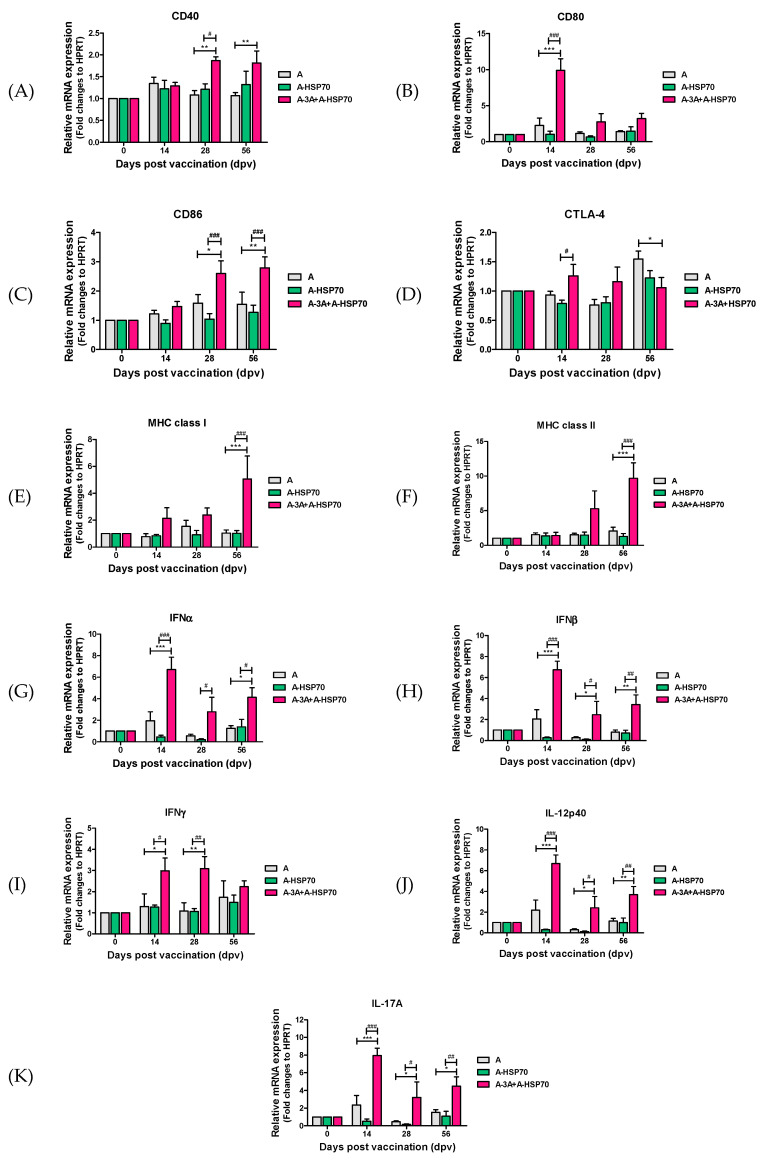
A-3A and A-HSP70 induced co-stimulatory molecule and cytokine gene expression in porcine PBMCs. Porcine peripheral blood mononuclear cells (PBMCs) isolated from the whole blood of vaccinated pigs (*n* = 5/group) individually, as described in Figure 4, were used for the qRT-PCR assay. Gene expression levels were normalized to *hprt* levels and are presented as a relative ratio compared to control levels. (**A**–**K**) represent: gene expression levels of (**A**) CD40; (**B**) CD80; (**C**) CD86; (**D**) CTLA-4; (**E**) MHC class I; (**F**) MHC class II; (**G**) IFNα; (**H**) IFNβ; (**I**) IFNγ; (**J**) IL-12p40; (**K**) IL-17A in porcine PBMCs were analyzed. The data represent the means ± SEM of triplicate measurements (*n* = 5/group). Statistical analyses were performed using one-way ANOVA with Tukey’s post-hoc test. *^, #^
*p <* 0.05; **^, ##^
*p <* 0.01; ***^, ###^
*p* < 0.001.

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
