# Peer review of "Advanced Foot-And-Mouth Disease Vaccine Platform for Stimulation of Simultaneous Cellular and Humoral Immune Responses"

_vaccines, 2020, doi:10.3390/vaccines8020254_

Round 1
Reviewer 1 Report
Minor comments
- I realized that the space in the abstract is very limited, but the first sentence would be greatly improved if the authors could mention one or two limitations.
- The first sentence in the abstract is too long and hard to read please break it into two or three sentences.
- Line 69 – This sentence should have “the” instead of “these”.
- Lines 69-72. This paragraph is one sentence. Please divide it into two or more sentences.
- Lines 74-78. This paragraph is somewhat confusing. First, they talk about vaccines A-3A and A-HSP70 and then using the antigens generated from these vaccines as new vaccines. It is not clear how these antigens were generated; this detail is important. Please add a very short description of what you meant.
- Line 134. I believe that the proper term will be “transfection” and not “transduction”.
- Line 217. Were the animals kept in isolation or just contained?
- Lines 291 to 296. This paragraph is almost only one sentence which is very hard to read. Please split it into several sentences.
Mayor comments
- In section 2.3 they describe how they made “inactivated viral antigens”. From this section I gather that they infected cells with the mutated viruses produced in section 2.2, then 24 hours post-infection, purified and neutralized. The term inactivated viral antigens is confusing. In fact, I believe is the wrong term, these are not inactivated viral antigens; these are inactivated viruses. I am aware that there are companies and groups calling them like this, but antigen is used to describe a protein or portion of a protein that interacts with an antibody. I´m afraid that the authors want to use the term “viral antigen” to avoid the real term that is “inactivated virus”, because we are all aware of the possible defects that a vaccine with inactivated viruses could have. Therefore, I strongly suggest replacing the terms inactivated viral antigens for inactivated viruses presenting the A-3A and A-HSP70 antigens. This becomes more apparent in section 2.4.2; here they talk about antigens acting as masters seed virus. This terminology is wrong. Antigens do not give rise to viruses. I am afraid that the authors are confused about some basic virology or terminology, please correct this as it is all over the article.
- Section 2.4.2 This section is confusing. First, they say that the animals were vaccinated at day 0 and challenged at day 7, but in the following paragraph they say that vaccination was performed twice. Something in the explanation is missing. I guess that the second paragraph refers only to pigs, so this needs to be clarified. Also, it is not clear if the protocol to collect sample is the same for pigs and for mice, please clarify. Finally, the second paragraph of this section is quite detailed about the types of vaccines and their composition, however, that is not the case for the third paragraph. I suggest to completely redo this section to provide clarity.
- Section 2.5.1 During the text the term “emulsion-free” was used, but I don´t think that in the introduction the importance of the presence or absence of oil was mentioned. I think that the importance of their experiments could be highlighted by explaining why finding an emulsion-free vaccine is important.
- I am confused by sections 2.5.2 and 2.4.2, it seems like there is a typo and section 2.4.2 was only intended for mice and 2.5.2 for pigs, but section 2.4.2 has information about the pigs. Also, it would be good the titles of these sections were consistent about terminology (vaccination or immunization) but specific enough we can know to which animal model they are referring to.
- Section 2.10 Was there any test to determine if the data was parametric or not? Please mention this.
- Figure 2. These experiments are missing one control group, non-infected and non-vaccinated. This group is important to understand the changes of BW. If they authors do have this group, please clarify it.
- Figure 2C and 2F. Is there any statistically significant difference in the weight of the animals?
- Figure 3B and C. These graphs are missing the statistical analysis (as it was done for the pigs). Would the authors explain why the 3B and 3C have almost no dispersion (there SD bars are not even showing) while the data for 4B and 4C shows a great deal of dispersion? Is this a problem related with the technique in pigs? Is it a response typical for pigs?, Please clarify this. Aslo, the figure legend of figures 2 and 3 do not indicate that any statistical test was performed. Please do them.
Author Response
Response to Reviewer 1 Comments and Suggestions
Minor comments
Thank you very much for your overall review.
We revised manuscript in depth per your recommendations.
Please see the response as follows and attachment (revised manuscript including supplementary materials).
1. I realized that the space in the abstract is very limited, but the first sentence would be greatly improved if the authors could mention one or two limitations.
Reviewer 1
Thank you for your kind comment.
As you recommended, we mentioned limitations of currently-available commercial FMD vaccines in line 14-15.
2. The first sentence in the abstract is too long and hard to read please break it into two or three sentences.
Reviewer 1
Thank you for your comment.
We divided the sentence in line 14-19 according your recommendation.
3. Line 69 – This sentence should have “the” instead of “these”.
Reviewer 1
Thank you for your comment.
We revised the sentence in line 73 according your suggestion.
4. Lines 69-72. This paragraph is one sentence. Please divide it into two or more sentences.
Reviewer 1
Thank you for your comment.
We divided and revised the sentence in line 73-78 according your comment.
5. Lines 74-78. This paragraph is somewhat confusing. First, they talk about vaccines A-3A and A-HSP70 and then using the antigens generated from these vaccines as new vaccines. It is not clear how these antigens were generated; this detail is important. Please add a very short description of what you meant.
Reviewer 1
Thank you for your comment.
We have added additional explanation in line 79-80 to address your point.
6. Line 134. I believe that the proper term will be “transfection” and not “transduction”.
Reviewer 1
Thank you for your critical comment.
We corrected 'transduction' to 'transfection' in line 143.
7. Line 217. Were the animals kept in isolation or just contained?
Reviewer 1
Thank you for your comment.
We modified the sentence in line 236 according your suggestion.
8. Lines 291 to 296. This paragraph is almost only one sentence which is very hard to read. Please split it into several sentences.
Reviewer 1
Thank you for your comment.
We modified and revised the sentence in line 326-332 according your comment.
Major comments
1. In section 2.3 they describe how they made “inactivated viral antigens”. From this section I gather that they infected cells with the mutated viruses produced in section 2.2, then 24 hours post-infection, purified and neutralized. The term inactivated viral antigens is confusing. In fact, I believe is the wrong term, these are not inactivated viral antigens; these are inactivated viruses. I am aware that there are companies and groups calling them like this, but antigen is used to describe a protein or portion of a protein that interacts with an antibody. I´m afraid that the authors want to use the term “viral antigen” to avoid the real term that is “inactivated virus”, because we are all aware of the possible defects that a vaccine with inactivated viruses could have. Therefore, I strongly suggest replacing the terms inactivated viral antigens for inactivated viruses presenting the A-3A and A-HSP70 antigens. This becomes more apparent in section 2.4.2; here they talk about antigens acting as masters seed virus. This terminology is wrong. Antigens do not give rise to viruses. I am afraid that the authors are confused about some basic virology or terminology, please correct this as it is all over the article.
Reviewer 1
Thank you for your critical comment.
We revised manuscript according your opinion (line 157-159).
We corrected this sentence (line 177).
2. Section 2.4.2 This section is confusing. First, they say that the animals were vaccinated at day 0 and challenged at day 7, but in the following paragraph they say that vaccination was performed twice. Something in the explanation is missing. I guess that the second paragraph refers only to pigs, so this needs to be clarified. Also, it is not clear if the protocol to collect sample is the same for pigs and for mice, please clarify. Finally, the second paragraph of this section is quite detailed about the types of vaccines and their composition, however, that is not the case for the third paragraph. I suggest to completely redo this section to provide clarity.
Reviewer 1
Thank you for your detailed comment.
We are sorry for the confusion arising from an unclear explanation.
2.4. section describes mouse experiments, not pig experiments. We clarified the title in line 183.
The sentence was modified and paragraph was divided to avoid confusion in 2.4.2 (Line 194-205).
Reviewer 1
Thank you for your comment.
We clarified the sentence in line 195-205. Please refer to Fig. 2A.
As mentioned in the text, vaccine compositions used in the experiments were as follows: purified antigens isolated from A-3A and A-HSP70 (15 μg/dose/mL 1/10∼1/640 dose for pig), ISA 206 (50%, w/w), 10% Al(OH)3, and 15 μg/mouse Quil-A.
In this experiment, we monitored survival rate and body weight post challenge.
We clarified the negative control group in line 202-203.
Reviewer 1
Thank you for your comment.
Mice were vaccinated on day 0 and day 35 to evaluate the long-term immunity of antigens derived from A-3A and A-HSP70 (line 206-220).
Please refer to Fig. 3A, which shows the experimental strategy.
As mentioned in the text, vaccine compositions used in the experiments were as follows: A-3A, A-HSP70, or A-3A+A-HSP70 antigens (15 μg/dose/mL, A-3A or A-HSP 1/10 dose for pig, A-3A 1/20 dose+A-HSP70 1/20 dose for pig), without (w/o) oil emulsion, 10% Al(OH)3, and 15 μg/mouse Quil-A.
In this experiment, blood samples were collected at 0, 7, 14, 28, 35, 56, and 84 dpv to measure the antibody titers and VN titers from the sera.
We clarified the negative control group in line 214-216.
3. Section 2.5.1 During the text the term “emulsion-free” was used, but I don´t think that in the introduction the importance of the presence or absence of oil was mentioned. I think that the importance of their experiments could be highlighted by explaining why finding an emulsion-free vaccine is important.
Reviewer 1
Thank you for your critical comment.
To improve the shortcomings of current commercially-available vaccines and enhance vaccine efficacy, our group is performing the research with two strategies. The first is the development of immunopotent vaccine strains, and the second is the development of non-oil adjuvants.
FMD vaccines generally contain oil-based adjuvants to improve vaccine efficacy. However, these adjuvants cause side effects due to oil in the vaccination site (Line 68-70). To address these problems, it is necessary to develop and apply non-oil adjuvants, including emulsions and immunostimulants.
Lines 73-78 indicate the disadvantages of oil-based adjuvants, especially oil emulsions, and why non-oil adjuvant (oil-free emulsion) experiments are needed.
We also described the importance of oil emulsion-free vaccines in lines 580-582.
Reviewer 1
Thank you for your comment.
We added an additional explanation explaining why non-oil emulsion (emulsion-free) vaccines are important (line 73-78).
Reviewer 1
Thank you for your critical comment in major comment #3.
We described the importance of oil emulsion-free vaccines in lines 580-582.
4. I am confused by sections 2.5.2 and 2.4.2, it seems like there is a typo and section 2.4.2 was only intended for mice and 2.5.2 for pigs, but section 2.4.2 has information about the pigs. Also, it would be good the titles of these sections were consistent about terminology (vaccination or immunization) but specific enough we can know to which animal model they are referring to.
Reviewer 1
Thank you for your comment.
We apologize for the confusion caused by an unclear explanation.
As shown in line 222, section 2.5 describes target animal (pigs) experiments, not mice experiments.
5. Section 2.10 Was there any test to determine if the data was parametric or not? Please mention this.
Reviewer 1
Thank you for your comment.
We described additional statistical tests according your comment (line 308-312).
6. Figure 2. These experiments are missing one control group, non-infected and non-vaccinated. This group is important to understand the changes of BW. If they authors do have this group, please clarify it.
Reviewer 1
Thank you for your meaningful comment.
In Fig. 2, the NC (negative control) group represents the group challenged with FMDV 7 days after receiving PBS instead of the vaccine.
We added the description of the NC group in line 202-203 and line 214-216.
Mice in the naïve control group (unvaccinated and unchallenged group) showed complete survival and normal increase in BW up to 7 dpc (data not shown). Please refer to line 361-363.
Reviewer 1
Thank you for your comment.
We clarified the description of the negative control group in line 202-203.
Reviewer 1
Thank you for your comment.
We clarified the description of the negative control group in line 202-203.
Reviewer 1
Thank you for your comment.
We clarified the description of the negative control group in line 202-203.
Reviewer 1
Thank you for your comment.
We clarified the description of the negative control group in line 214-216.
7. Figure 2C and 2F. Is there any statistically significant difference in the weight of the animals?
Reviewer 1
Thank you for your critical comment.
We added statistical analysis in Supplementary Table 3A and Table 3B. This is mentioned in Fig. 2 legend (line 377).
8. Figure 3B and C. These graphs are missing the statistical analysis (as it was done for the pigs). Would the authors explain why the 3B and 3C have almost no dispersion (there SD bars are not even showing) while the data for 4B and 4C shows a great deal of dispersion? Is this a problem related with the technique in pigs? Is it a response typical for pigs? Please clarify this. Aslo, the figure legend of figures 2 and 3 do not indicate that any statistical test was performed. Please do them.
Reviewer 1
Thank you for your comment.
The data represent the means±SEM of triplicate measurement (n=5/group).
In the mice experiment using SPF animals, the SEM between individuals was low, and thus the error bar is barely visible. On the other hand, in the case of pigs, the FMD vaccine-mediated immune response has a large difference among individuals, and the SEM value was high because field animals were used in this experiment. We revised the Fig. 3B, 3C by changing the graph type from 'broken line' to 'dot plot'.
We revised the figure by adding statistical significance in Fig. 3B and 3C, and added statistical analysis in Supplementary Table 4A and 4B. This is mentioned in Fig. 3 legend (line 445-446).

Reviewer 2 Report
The authors of this study report the development of potential vaccines strains able to induce increased/improved cellular and adaptive immune responses than those induced by conventional vaccine strains. This is a well written manuscript and the methods, results and conclusions are sound.
My main observation to this manuscript is on the description of the methods. This section is not complete and needs to be improved . For several experiments de descriptions of the methodology are better or clearer explained in the results section. I would recommend to revise this. In addition more information describing housing and welfare conditions applied to the animals (3Rs) during experimentation is required: eg. adaptation period for mice and pigs (used in containment) before starting the experiments, housing/enrichment, etc. This is important if other scientist would like to replicate this study. I suggest the authors look at the ARRIVE guidelines.
Minor observations:
L36. FMD does not induce high fatality in their natural hosts. This is a wrong statement even if it has a reference, it does not mean is correct! Mortality is not a main characteristic of FMD and it is dependent on the host. Mortality may happen in young animals.
L201: In the results section, it is also reported challenge of one group at dpv 168. This is missing in this section.
L209: please explain what are the treatments in the two groups. I assume that one group treatment is A-HSP70 and the second group? The Sup. Figure mentions a A (positive control). Please mention this in the main text and explain what do you mean with "positive control"
L217: Please clarify animals from which experiment where kept in containment. The experiment described in L205 - L210 are mentioned as "field" experiments. This may cause some confusion.
L301 - 307: You refer here to Supp Fig. 2 but I think you are talking about Supp Fig 1?
L371 - 372: What about A-3A alone? Looking at Fig3 A-3A has similar antibody levels to those reached by A-HSP70. Why is one mentioned and the other ignored?
L373-378: Not clear in the main text the referencing to the figures. Please clarify which figures refer to the groups challenged at 84 dpv and which ones to the 168 dpv.
L389: Please revise: "...compared with the A alone group and the positive control group" I think the A alone and the positive control group are the same?
L438: "but was slightly higher in the group A..." Delete "slightly". One gets the impression of positive bias in the reporting of results. Particularly if you see the size of the difference between A-3A+A-HSP70 vs A at 14 dpv. You did not report this difference as "slightly".
L422: revise "A-HSP" for consistency use "A-HSP70"
L440: "The gene expression levels of cytokines were higher in the A-3A+A-HSP70 and A-HSP70 alone groups compared with the A alone group" Looking at the figures this statement does not apply to the A-HSP70 alone.
Figure 5: Please present two figures per row. e.g A and B, C and D, etc
Author Response
Response to Reviewer 2 Comments and Suggestions
The authors of this study report the development of potential vaccines strains able to induce increased/improved cellular and adaptive immune responses than those induced by conventional vaccine strains. This is a well written manuscript and the methods, results and conclusions are sound.
My main observation to this manuscript is on the description of the methods. This section is not complete and needs to be improved . For several experiments de descriptions of the methodology are better or clearer explained in the results section. I would recommend to revise this. In addition more information describing housing and welfare conditions applied to the animals (3Rs) during experimentation is required: eg. adaptation period for mice and pigs (used in containment) before starting the experiments, housing/enrichment, etc. This is important if other scientist would like to replicate this study. I suggest the authors look at the ARRIVE guidelines.
Thank you very much for your overall review.
We revised manuscript in depth per your recommendations.
Please see the response as follows and attachment (revised manuscript including supplementary materials).
Reviewer 2
Thank you for your nice comment.
We mentioned the animal adaptation period, food availability, and housing conditions (line 187-191).
Reviewer 2
Thank you for your nice comment.
We mentioned the animal adaptation period, food availability, and housing conditions (line 236-239).
Minor observations:
1. L36. FMD does not induce high fatality in their natural hosts. This is a wrong statement even if it has a reference, it does not mean is correct! Mortality is not a main characteristic of FMD and it is dependent on the host. Mortality may happen in young animals.
Reviewer 2
Thank you for your crucial comment.
We revised the sentence in line 39-40.
2. L201: In the results section, it is also reported challenge of one group at dpv 168. This is missing in this section.
Reviewer 2
Thank you for your thorough review.
We corrected the sentence in line 216.
In addition, A (backbone strain) antigen was used as positive control in this experiment.
We revised the sentence according to your recommendation (line 211).
3. L209: please explain what are the treatments in the two groups. I assume that one group treatment is A-HSP70 and the second group? The Sup. Figure mentions a A (positive control). Please mention this in the main text and explain what do you mean with "positive control"
Reviewer 2
Thank you for your critical comment.
A (backbone strain) antigen was used as positive control in this experiment.
We clarified each group in lines 228-229.
4. L217: Please clarify animals from which experiment where kept in containment. The experiment described in L205 - L210 are mentioned as "field" experiments. This may cause some confusion.
Reviewer 2
Thank you for your comment.
For the target animal experiment, we used field animals (pigs purchased in the farm), not SPF animals. Therefore, we described this as a field experiment. The animals were isolated in closed containments (ABSL3) during the study as described in line 236.
We revised this sentence for better understanding (line 236).
Reviewer 2
Thank you for your critical comment.
We corrected 'field experiment' to 'target animal experiment' in line 225-226.
5. L301 - 307: You refer here to Supp Fig. 2 but I think you are talking about Supp Fig 1?
Reviewer 2
Thank you for your comment.
We revised the text according to your recommendation (line 340-342).
6. L371 - 372: What about A-3A alone? Looking at Fig3 A-3A has similar antibody levels to those reached by A-HSP70. Why is one mentioned and the other ignored?
Reviewer 2
Thank you for your comment.
This sentence means that the antibody titer was high in the A-3A alone group and the A-3A+A-HSP70 co-administration group.
We revised the text per your recommendation (line 410-411).
7. L373-378: Not clear in the main text the referencing to the figures. Please clarify which figures refer to the groups challenged at 84 dpv and which ones to the 168 dpv.
Reviewer 2
Thank you for your comment.
We clarified the results (Fig. 3D-3G) of the challenge experiments at 84 dpv and 168 dpv in lines 414-418.
8. L389: Please revise: "...compared with the A alone group and the positive control group" I think the A alone and the positive control group are the same?
Reviewer 2
Thank you for your comment.
We revised the sentence and deleted "and the positive control group" in line 438.
9. L438: "but was slightly higher in the group A..." Delete "slightly". One gets the impression of positive bias in the reporting of results. Particularly if you see the size of the difference between A-3A+A-HSP70 vs A at 14 dpv. You did not report this difference as "slightly".
Reviewer 2
Thank you for your comment.
We revised the sentence an deleted "slightly" in line 477.
10. L422: revise "A-HSP" for consistency use "A-HSP70"
Reviewer 2
Thank you for your comment.
We modified 'A-HSP' to 'A-HSP70' in line 462.
11. L440: "The gene expression levels of cytokines were higher in the A-3A+A-HSP70 and A-HSP70 alone groups compared with the A alone group" Looking at the figures this statement does not apply to the A-HSP70 alone.
Reviewer 2
Thank you for your comment.
We revised the sentence in lines 479-480.
12. Figure 5: Please present two figures per row. e.g A and B, C and D, etc
Reviewer 2
Thank you for your comment.
We rearranged Fig. 5 according to your recommendations.

Reviewer 3 Report
In the submitted manuscript Lee et al. developed FMDV inactivated recombinant vaccine candidates by insertion of either sequence of FMDV 3A epitope or sequence of HSP70 epitope into VP1 region of a FMDV virus based on O1-Manisa backbone containing capsid of A22/Iraq/24/64 isolate. Authors test both vaccine candidates and a combination of both vaccines in mice for induction of humoral immunity and protection from a challenge virus. In target species i.e. pigs authors tested the vaccine candidate containing the epitope of HSP70 and a combination of two vaccine candidates for humoral immune response and for mRNA levels of number of immune markers from pig PBMCs.
Overall authors carried out an extensive study and most figures were clearly presented.
1) There exists an inconsistency when describing insertion of 3A and HSP70 epitopes into recombinant FMDV construct: authors state that epitopes were inserted between 4380 and 4381 nts of the recombinant FMDV construct; Fig. 1 suggests that authors replaced aa 152 – 166 and 152 -171 of VP1 of the recombinant FMDV construct with 3A epitope and HSP70 epitope, respectively; while Table S1 suggests that comparing to the recombinant FMDV construct, constructs containing 3A or HSP70 epitope are missing 60 nts which are downstream of VP1 nucleotide number 4380 of the original recombinant FMDV construct. Without clear description how the recombinant vaccine candidates were generated it is impossible to take any decision on the rest of the manuscript.
3) When presenting TEM data an image of a control virus (i.e. FMDV-A ‘only’) should be included.
2) Material and method section 2.1 is not clearly written and requires rewriting.
Sections describing vaccine and animal studies contain many repeats and require logical structuring.
It appears that only for some animal studies authors state usage of oil-free vaccines; does it mean that for other studies authors used oil-based vaccines? Please state for each experiment in figure legend.
3) Authors do not appear to explain logic behind their experimental strategy before describing their results. They clearly based insertion of epitopes into VP1 on Seago et al. 2012 (with certain modifications, which also should be explained) and selected 3A and HSP70 epitopes based on previously published work. Reason for the experimental design supported, where appropriate, by references should be included.
4) 1.5 ug of antigen seems to be very high dose for mice vaccination.
5) I personally would be very careful when interpreting results of mRNA expression of cell surface markers. Even if clear increase of mRNA levels would be observed, it still doesn’t indicate their increased expression on cell surface.
There is some previous evidence that IL-6 and IL-8 increased serum levels may indicate good protection induced by a vaccine. These cytokines might be worth testing.
6) Thank you for a very extensive study; did authors actually tested 3A+HSP70 vaccine candidate against a challenge virus in pigs? I personally think that even if there was no protection, this is still a valid result and would be good to include it.
7) Calling a vaccine candidate successful in mice when only a very high dose is used appears as over-interpretation of results. A 1.5 ug (1/10th of a dose used for pig) seems huge amount since mice probably weight ~20 g while pigs probably ~20 kg.
8) If authors conducted preliminary tests of the A-3A vaccine candidate in pigs (as it was done for A-HSP70 in supplementary figure 2) and actually tested the A-HSP70 and A-3A+A-HSP70 against virus challenge in pigs, it would be very beneficial for the scientific community to include these results even if authors observed no difference comparing to the control vaccine.
Author Response
Response to Reviewer 3 Comments and Suggestions
In the submitted manuscript Lee et al. developed FMDV inactivated recombinant vaccine candidates by insertion of either sequence of FMDV 3A epitope or sequence of HSP70 epitope into VP1 region of a FMDV virus based on O1-Manisa backbone containing capsid of A22/Iraq/24/64 isolate. Authors test both vaccine candidates and a combination of both vaccines in mice for induction of humoral immunity and protection from a challenge virus. In target species i.e. pigs authors tested the vaccine candidate containing the epitope of HSP70 and a combination of two vaccine candidates for humoral immune response and for mRNA levels of number of immune markers from pig PBMCs.
Overall authors carried out an extensive study and most figures were clearly presented.
Thank you very much for your overall review.
We revised manuscript in depth per your recommendations.
Please see the response as follows and attachment (revised manuscript including supplementary materials).
1) There exists an inconsistency when describing insertion of 3A and HSP70 epitopes into recombinant FMDV construct: authors state that epitopes were inserted between 4380 and 4381 nts of the recombinant FMDV construct; Fig. 1 suggests that authors replaced aa 152 – 166 and 152 -171 of VP1 of the recombinant FMDV construct with 3A epitope and HSP70 epitope, respectively; while Table S1 suggests that comparing to the recombinant FMDV construct, constructs containing 3A or HSP70 epitope are missing 60 nts which are downstream of VP1 nucleotide number 4380 of the original recombinant FMDV construct. Without clear description how the recombinant vaccine candidates were generated it is impossible to take any decision on the rest of the manuscript.
Reviewer 3
Thank you for your critical comment.
We clarified the method for construction the A-3A and A-HSP70 in Fig. 1 and Table S1 according to your recommendations.
Please refer to lines 89-142 and Table S1.
2) When presenting TEM data an image of a control virus (i.e. FMDV-A ‘only’) should be included.
Reviewer 3
Thank you for your comment.
We added the results of the PBM rapid kit and TEM data for FMDV A in supplementary Fig. 1.
3) Material and method section 2.1 is not clearly written and requires rewriting.
Sections describing vaccine and animal studies contain many repeats and require logical structuring.
It appears that only for some animal studies authors state usage of oil-free vaccines; does it mean that for other studies authors used oil-based vaccines? Please state for each experiment in figure legend.
Reviewer 3
Thank you for your comment.
We revised the Materials and Methods section 2.1 and corrected Table S1, per your recommendations.
For mouse and pig experiments, the vaccine composition (e.g., emulsion-containing or emulsion-free, administration of each antigen or combination from an immunopotent FMD vaccine strain, etc.) and research strategy (short-term, mid-term, long-term immunity, etc.) are different. Therefore, a detailed explanation of each experiment was required for better understanding.
Based on your recommendations, we clarified the use of oil-based or oil-free vaccines in each figure legend.
Please refer to lines 370, 433, 436, and 449-451.
4) Authors do not appear to explain logic behind their experimental strategy before describing their results. They clearly based insertion of epitopes into VP1 on Seago et al. 2012 (with certain modifications, which also should be explained) and selected 3A and HSP70 epitopes based on previously published work. Reason for the experimental design supported, where appropriate, by references should be included.
Reviewer 3
Thank you for your critical comment.
We mentioned the need for the development of immunopotent FMDV vaccine strains in the Introduction, and the background for selecting 3A and HSP70 as target molecules and inserting them on the FMDV surface are explained in the Results section (3.1) and Discussion.
Our goal in this paper is different from Seago et al. (2012). References to T cell epitope such as 3A have already been described in this manuscript. We revised the differences in goals between other reports and our study in lines 561-564.
5) 1.5 ug of antigen seems to be very high dose for mice vaccination.
Reviewer 3
Thank you for your definitive comment.
To confirm the immunogenicity and short-term immunity of purified antigens isolated from immunopotent FMDVs, A-3A and A-HSP70, the does response (PD50 value) in mice was evaluated (Fig. 2).
This experiment was performed using a vaccine containing an oil-based emulsion.
In the next experiment, the initial, mid-term, and long-term immunity were evaluated using an oil emulsion-free test vaccine (Fig. 3).
In the oil-emulsion free state, the antigen is rapidly released into the host, making it difficult to maintain a constantly high antibody titer. Therefore, the experiment was conducted using 1.5 μg antigen, which is 1/10 of the target animal (pig) dose. We will evaluate the induction of immune responses and maintenance of antibody titers at lower antigen concentrations an optimize the antigen dose to increase vaccine efficacy in further studies.
Please refer to lines 211-212.
6) I personally would be very careful when interpreting results of mRNA expression of cell surface markers. Even if clear increase of mRNA levels would be observed, it still doesn’t indicate their increased expression on cell surface.
There is some previous evidence that IL-6 and IL-8 increased serum levels may indicate good protection induced by a vaccine. These cytokines might be worth testing.
Thank you for your comment.
We fully agree your opinion.
Extracellular cytokine expression is more important than intracellular cytokine expression.
IL-6 and IL-8 expression levels are crucial in inducing adaptive immune responses and forming antibodies, and FMD vaccine-mediated IL-6 and IL-8 expression were confirmed by Cox et al. (Vaccine, 2003).
In a previous study, we also confirmed inactivated FMD virus (antigen)-mediated extracellular and intracellular murine, bovine, and porcine cytokine profiles and time kinetics by ELISA and RNA-seq, respectively. We described these results in two other manuscripts that are currently under review.
It is difficult to obtain various commercially-available porcine ELISA kits and FACS antibodies to detect porcine cytokines, chemokines, and co-stimulatory molecules. Therefore, studies of FMD vaccine-induced immune responses in pigs are limited. We will attempt to identify FMD vaccine-mediated cellular and humoral immune responses through various strategies in further studies.
Please refer to line 489.
7) Thank you for a very extensive study; did authors actually tested 3A+HSP70 vaccine candidate against a challenge virus in pigs? I personally think that even if there was no protection, this is still a valid result and would be good to include it.
Reviewer 3
Thank you for your critical comment.
We have not yet performed the challenge trial on pigs that were administrated test vaccine including A-3A+A-HSP70 antigen.
In addition to the A-3A and A-HSP70 mentioned in this study, other immunopotent FMDVs are being tested. We are planning a challenge experiment using these vaccines in the second half of this year.
Please refer to line 578.
8) Calling a vaccine candidate successful in mice when only a very high dose is used appears as over-interpretation of results. A 1.5 ug (1/10th of a dose used for pig) seems huge amount since mice probably weight ~20 g while pigs probably ~20 kg.
Reviewer 3
Thank you for your definitive comment.
Comment #3 and #8 are interpreted in the same context.
We summarized the changes based on your recommendations as follows:
To confirm the immunogenicity and short-term immunity of purified antigens isolated fro, immunopotent FMDVs, A-3A and A-HSP70, we evaluated the dose responses (PD50 value) in mice (Fig. 2).
This experiment was performed using a vaccine containing an oil-based emulsion.
In the next experiment, the initial, mid-term, and long-term immunity were evaluated using an oil emulsion-free test vaccine (Fig. 3).
In the oil-emulsion free state, the antigen is rapidly released into the host, making it difficult to maintain a constantly high antibody titer. Therefore, the experiment was conducted using 1.5 μg antigen, which is 1/10 of the target animal (pig) dose. We will evaluate the induction of immune responses and maintenance of antibody titers at lower antigen concentrations an optimize the antigen dose to increase vaccine efficacy in further studies.
Please refer to lines 211-212.
9) If authors conducted preliminary tests of the A-3A vaccine candidate in pigs (as it was done for A-HSP70 in supplementary figure 2) and actually tested the A-HSP70 and A-3A+A-HSP70 against virus challenge in pigs, it would be very beneficial for the scientific community to include these results even if authors observed no difference comparing to the control vaccine.
Reviewer 3
Thank you for your crucial comment.
We constructed A-HSP70, followed by A-3A.
FMD vaccines generally contain oil-based oil emulsion. Therefore, we performed an experiment (Supplementary Fig. 2) to compare recombinant FMD vaccine strains developed by our group or a commercially available vaccine with the immunopotent vaccine strain A-HSP70.
In case of A-3A, after confirming the immune response induced by the oil-based emulsion and oil emulsion-free test vaccine in mice, the experiment was performed in the oil-emulsion free state by co-administion with A-HSP70 in pigs (Fig. 4).
It is very challenging to observe the induction of immune responses in pigs using an oil emulsion-free vaccine.
The immune responses were higher in the A-3A+A-HSP70 co-administrated group than in the A-HSP70 alone administrated group in pigs (Fig. 4). Induction of the immune response in the A-3A alone administrated group in mice was stronger than the A-HSP70 alone-administrated group (Fig. 2 and Fig. 3). Based on these results, we expect that A-3A will elicit a potent immune response than A-HSP70 in pigs.
To prove this, we are planning a challenge experiment with FMDV after vaccination with A-3A alone (in the challenge experiment above mentioned in 8).
Please refer to lines 379-380.

Round 2
Reviewer 3 Report
Thank you authors for your replies.
Here are my comments:
1)
The description of the construction of A-3A and A-HSP70 infectious copy plasmids is still very confusing. The way I understand the article is that for A-3A a fragment of FMDV 3A sequence was inserted between 4830 and 4381 nt (which is in the VP1 encoding region) of the FMDV infectious copy plasmid, and for A-HSP70 a sequence of HSP70 was inserted in the same region of the FMDV infectious copy plasmid. But from the way section 2.1 is written and Fig. 1 is presented this is not clear. It seems that authors switch between referring to aa residues of the inserts and VP1 of the vector in such way that it is confusing for the reader. Also in lines 108 and 119 authors state that 3A and HSP70 epitopes were inserted in between 4380 and 4381 nt of the FMDV infectious copy plasmid, but then in lines 149-50 authors say that “The insertion of HSP70 into the amino acid sequence 152-171 (6th RGD amino acids) of the VP1” (similar is shown in Fig. 1), which is very confusing. If authors wish to show that they made a chimeric PCR fragment of VP1 and 3A/HSP-70 to be inserted into the FMDV infectious copy, then that should be clearly stated.
Here are some more specific examples:
In lines 104-112 when saying [my comments in square brackets] “With the plasmids prepared as described above (pOm-A22-P1), the T cell epitope 3A (GCAGCAATTGAATTCTTTGAGGGAATGGTGCATGACTCCATCAAG (SEQ ID No. 4), which corresponds to the amino acid residue sequence 152-166 (AAIEFFEGMVHDSIK (SEQ ID No. 5) [I assume you mean it corresponds to the aa sequence of the 3A], 6th RGD amino acid [but then the mentioned RGD doesn’t make sense]), was inserted between the 4380 and 4381 base sequence, and the above plasmid (300 ng/μL), primer 3A F (5’-GGAATGGTGCATGACTCCATCAAGGCGAGGGTCGCCGCTCAGCT-3’ (SEQ ID 110 No. 6)), 10 pmole/μL, 1 μL), and primer 3A R (5’- CTCAAAGAATTCAATTGCTGCCGCGAGAGGCCCTAGGTCGC-3’ (SEQ ID No. 7), 10 pmole/μL, 1 μL) were used.
In lines 114-24: “For A-HSP70, the heat shock protein (HSP) 70 sequence (CAACCGTCGGTGCAGATCCAGGTCTATCAGGGGGAGCGTGAGATCGCCGCGCACAACAAG (SEQ ID No. 9)), which corresponds to the amino acid residue sequence 152-171 (QPSVQIQVYQGEREIAAHNK (SEQ ID No. 10), 6th RGD amino acid) of the VP1 [is this aa sequence of HSP70 or VP1???; this is very confusing], was inserted between the base sequence 4380 and 4381 of SeqSEQ ID No. 3 for A-HSP70 [delete “for A-HSP70”].
For the description of construction of A-HSP70 it would also be useful to mention the accession number of the sequence were the insert comes from and the origin of the species.
For Fig. 1 under aa sequence of 3A and HSP70 fragments authors say xx-xx aa of the VP1 and in the legend they say “FMDV T cell epitope 3A [the amino acid residue sequence 152-166 (6th RGD amino acid) of the VP1] (A) and enhanced long-term immunity protein HSP70 [amino acid residue sequence 152-171 (6th RGD amino acid) of the VP1] (B) “: so is it sequence of 3A/HSP70 or VP1? It might be just a language issue, but it does need to be corrected.
2)
Lines 128-53: X ul of something was a correct phrase.
3)
Lines 353-55: thank you for explaining why 3A and HSP70 epitopes were selected. Please insert appropriate references at the end of the sentence.
4)
Section 3.1 Of course that the aim of authors’ study was different to the aim of Seago et al. 2012, nevertheless Seago et al. was the first manuscript showing that short epitopes can be inserted in that region of VP1 of FMDV resulting in a viable virus and therefore that manuscript should be cited in your paper when describing design of A-3A and A-HSP70. (In contrast lines 627-9 are unnecessary and I think authors just misunderstood my initial suggestion). Also it would be useful to specify in this section that the epitopes were inserted in the VP1 encoding region (despite that being described in materials and methods).
5)
Section 3.3 and statement in lines 390-94: Authors state that result support induced long term response but challenged mice at 7 dpv and that vaccines induce cellular responses no results for any immune responses are shown in Fig. 1. Long term immunity and antibody responses are shown in Fig 3, but not here. Therefore the statement in lines 390-94 should be changed to something like: the results from this challenge study are encouraging suggesting that both recombinant vaccines A-3A and A-HSP70 might grant protection from experimental infection in mice what is further validated later in this manuscript. The statement currently given, taking the data provided in Fig. 1 is an overstatement. Also, in Fig. 1 there is no comparison to A-only so at this stage we do not know whether the response to A-3A and/or A-HSP70 is specific.
6)
Supplementary Fig 1: there is only a title and no legend; sort description of what A-F shows would be beneficial.
7)
Lines 410-11: Authors call A-HSP70 an “enhanced long-term immunity FMD vaccine strain “before even proving these attributes; I think A-HSP70 vaccine strain would be a sufficient name.
8)
Lines 441-43 should say “higher than A-only control” not high antibody titres especially that these titres are not above the threshold.
9)
Fig. 5 legend (lines 533-4) “The data represent the means ± SEM of triplicate measurements (n = 3/group)” does it mean that the PCR was conducted three times or PBMCs from three animals per group was tested (if the latter why PBMCs from all 5 animals were not tested). Please correct accordingly.
11)
Line 574: authors tested the epitopes in the manuscript, not identified them, please correct.
12)
578 – please provide reference.
13)
Lines 21, 639 and 697: while it is correct to say that the tested vaccine (especially mixture of A-3A+A-HPS70) induces a robust humoral responses, I think that there are insufficient data showing a robust cellular response to be able to claim it. While shown qPCR results are encouraging, suggesting that the vaccines might in fact induce cellular responses, this need to be further tested. Please revise statements in the mentioned lines to reflect my comment. Similarly, please reflect that in lines 672-3 since qPCR results suggest that cellular immune response might be activated, but they do not confirm that is this case (further studies are required to verify cellular immune responses).
Author Response
1)
The description of the construction of A-3A and A-HSP70 infectious copy plasmids is still very confusing. The way I understand the article is that for A-3A a fragment of FMDV 3A sequence was inserted between 4830 and 4381 nt (which is in the VP1 encoding region) of the FMDV infectious copy plasmid, and for A-HSP70 a sequence of HSP70 was inserted in the same region of the FMDV infectious copy plasmid. But from the way section 2.1 is written and Fig. 1 is presented this is not clear. It seems that authors switch between referring to aa residues of the inserts and VP1 of the vector in such way that it is confusing for the reader. Also in lines 108 and 119 authors state that 3A and HSP70 epitopes were inserted in between 4380 and 4381 nt of the FMDV infectious copy plasmid, but then in lines 149-50 authors say that “The insertion of HSP70 into the amino acid sequence 152-171 (6th RGD amino acids) of the VP1” (similar is shown in Fig. 1), which is very confusing. If authors wish to show that they made a chimeric PCR fragment of VP1 and 3A/HSP-70 to be inserted into the FMDV infectious copy, then that should be clearly stated.
Here are some more specific examples:
In lines 104-112 when saying [my comments in square brackets] “With the plasmids prepared as described above (pOm-A22-P1), the T cell epitope 3A (GCAGCAATTGAATTCTTTGAGGGAATGGTGCATGACTCCATCAAG (SEQ ID No. 4), which corresponds to the amino acid residue sequence 152-166 (AAIEFFEGMVHDSIK (SEQ ID No. 5) [I assume you mean it corresponds to the aa sequence of the 3A], 6th RGD amino acid [but then the mentioned RGD doesn’t make sense]), was inserted between the 4380 and 4381 base sequence, and the above plasmid (300 ng/μL), primer 3A F (5’-GGAATGGTGCATGACTCCATCAAGGCGAGGGTCGCCGCTCAGCT-3’ (SEQ ID 110 No. 6)), 10 pmole/μL, 1 μL), and primer 3A R (5’- CTCAAAGAATTCAATTGCTGCCGCGAGAGGCCCTAGGTCGC-3’ (SEQ ID No. 7), 10 pmole/μL, 1 μL) were used.
In lines 114-24: “For A-HSP70, the heat shock protein (HSP) 70 sequence (CAACCGTCGGTGCAGATCCAGGTCTATCAGGGGGAGCGTGAGATCGCCGCGCACAACAAG (SEQ ID No. 9)), which corresponds to the amino acid residue sequence 152-171 (QPSVQIQVYQGEREIAAHNK (SEQ ID No. 10), 6th RGD amino acid) of the VP1 [is this aa sequence of HSP70 or VP1???; this is very confusing], was inserted between the base sequence 4380 and 4381 of SeqSEQ ID No. 3 for A-HSP70 [delete “for A-HSP70”].
For the description of construction of A-HSP70 it would also be useful to mention the accession number of the sequence were the insert comes from and the origin of the species.
For Fig. 1 under aa sequence of 3A and HSP70 fragments authors say xx-xx aa of the VP1 and in the legend they say “FMDV T cell epitope 3A [the amino acid residue sequence 152-166 (6th RGD amino acid) of the VP1] (A) and enhanced long-term immunity protein HSP70 [amino acid residue sequence 152-171 (6th RGD amino acid) of the VP1] (B) “: so is it sequence of 3A/HSP70 or VP1? It might be just a language issue, but it does need to be corrected.
Reviewer 3
Thank you for your critical comments.
We have made revision to clarify the 'recombinant plasmid preparation' section, and the reference and GeneBank Accession No are clearly marked , therefore supplementary Table 1 has been removed to avoid further reader's confusion in lines 89-112.
We revised the sentence to avoid confusion (lines 133-136).
We have clearly modified the schematic diagrams (Fig. 1A, 1B and Fig. 1 legend) for the construction of A-3A and A-HSP70 (lines 146-151).
2)
Lines 128-53: X ul of something was a correct phrase.
Reviewer 3
Thank you for your detailed comment.
We revised the paragraph as your recommendation (lines 114-138).
3)
Lines 353-55: thank you for explaining why 3A and HSP70 epitopes were selected. Please insert appropriate references at the end of the sentence.
Reviewer 3
Thank you for your meaningful comment.
We inserted the reference at the end of the sentence according to your recommendation in line 328.
4)
Section 3.1 Of course that the aim of authors’ study was different to the aim of Seago et al. 2012, nevertheless Seago et al. was the first manuscript showing that short epitopes can be inserted in that region of VP1 of FMDV resulting in a viable virus and therefore that manuscript should be cited in your paper when describing design of A-3A and A-HSP70. (In contrast lines 627-9 are unnecessary and I think authors just misunderstood my initial suggestion). Also it would be useful to specify in this section that the epitopes were inserted in the VP1 encoding region (despite that being described in materials and methods).
Reviewer 3
Thank you for your critical comment.
We cited the reference of Seago et al. (2012) according to your recommendation in lines 319-321.
We have removed the misinterpreted section in lines 561-564 related to the paper Seago et al. (2012), and the deleted contents are as follows.
"There are reports on the development of 3A-epitope , HA- and FLAG-tagged FMDVs by reverse genetics (65)(66). These tagged FMDVs were developed to provide alternatives to vaccine enrichment and purification, and provide a unique tool for FMDV research, rather than the perspective of developing immunopotent FMD vaccine strains."
We revised the sentence to clarify the insertion of epitope into VP1 encoding region in line 328-330.
5)
Section 3.3 and statement in lines 390-94: Authors state that result support induced long term response but challenged mice at 7 dpv and that vaccines induce cellular responses no results for any immune responses are shown in Fig. 1. Long term immunity and antibody responses are shown in Fig 3, but not here. Therefore the statement in lines 390-94 should be changed to something like: the results from this challenge study are encouraging suggesting that both recombinant vaccines A-3A and A-HSP70 might grant protection from experimental infection in mice what is further validated later in this manuscript. The statement currently given, taking the data provided in Fig. 1 is an overstatement. Also, in Fig. 1 there is no comparison to A-only so at this stage we do not know whether the response to A-3A and/or A-HSP70 is specific.
Reviewer 3
Thank you for your thorough review.
We revised text in lines 360-363 per your recommendation.
6)
Supplementary Fig 1: there is only a title and no legend; sort description of what A-F shows would be beneficial.
Reviewer 3
Thank you for your crucial comment.
We revised the Supplementary Fig. 1 legend according to your recommendation.
7)
Lines 410-11: Authors call A-HSP70 an “enhanced long-term immunity FMD vaccine strain “before even proving these attributes; I think A-HSP70 vaccine strain would be a sufficient name.
Reviewer 3
Thank you for your comment.
We revised the sentence 'enhanced long-term immunity FMD vaccine strain' to 'A-HSP70' in line 378.
8)
Lines 441-43 should say “higher than A-only control” not high antibody titres especially that these titres are not above the threshold.
Reviewer 3
Thank you for your comment.
We revised the sentence according to your recommendation in line 427.
9)
Fig. 5 legend (lines 533-4) “The data represent the means ± SEM of triplicate measurements (n = 3/group)” does it mean that the PCR was conducted three times or PBMCs from three animals per group was tested (if the latter why PBMCs from all 5 animals were not tested). Please correct accordingly.
Reviewer 3
Thank you for your critical comment.
PBMCs were individually isolated from the whole blood of 5 animals per group, and qPCR was performed triplicate for each sample.
We clarified the Fig. 5 legend in lines 488, and 492.
10)
Line 574: authors tested the epitopes in the manuscript, not identified them, please correct.
Reviewer 3
Thank you for your comment.
We revised the sentence according to your recommendation (line 521).
11)
578 – please provide reference.
Reviewer 3
Thank you for your comment.
We cited the reference per your recommendation in line 525.
12)
Lines 21, 639 and 697: while it is correct to say that the tested vaccine (especially mixture of A-3A+A-HPS70) induces a robust humoral responses, I think that there are insufficient data showing a robust cellular response to be able to claim it. While shown qPCR results are encouraging, suggesting that the vaccines might in fact induce cellular responses, this need to be further tested. Please revise statements in the mentioned lines to reflect my comment. Similarly, please reflect that in lines 672-3 since qPCR results suggest that cellular immune response might be activated, but they do not confirm that is this case (further studies are required to verify cellular immune responses).
Reviewer 3
Thank you for your critical comment.
We revised the sentence to reflect your opinion in lines 565-566, and 596-598.
Based on the background that 3A is T cell epitope, we interpret that A-3A induced T cell-mediated cellular immune response. Although qPCR results are not sufficient to verify A-3A induced cellular immunity, we propose that the expression of a vaccine-induced co-stimulatory molecules and cytokines containing an immunopotent FMD antigen can be induced by cellular immune response, and we will elucidate this in further studies.
